# A metagenomic analysis of the phase 2 *Anopheles gambiae* 1000 genomes dataset reveals a wide diversity of cobionts associated with field collected mosquitoes
Andrzej Pastusiak[1], Michael R. Reddy [1 ✉], Xiaoji Chen[1], Isaiah Hoyer[1], Jack Dorman[2], Mary E. Gebhardt[3], Giovanna Carpi [2], Douglas E. Norris [3], James M. Pipas[4] & Ethan K. Jackson[1]

The *Anopheles gambiae* 1000 Genomes (Ag1000G) Consortium previously utilized deep sequencing methods to catalogue genetic diversity across African *An. gambiae* populations. We analyzed the complete datasets of 1142 individually sequenced mosquitoes through Microsoft Premonition's Bayesian mixture model based (BMM) metagenomics pipeline. All specimens were confirmed as either *An. gambiae* sensu stricto (s.s.) or *An. coluzzii* with a high degree of confidence ( > 98% identity to reference). *Homo sapiens* DNA was identified in all specimens indicating contamination may have occurred either at the time of specimen collection, preparation and/or sequencing. We found evidence of vertebrate hosts in 162 specimens. 59 specimens contained validated *Plasmodium falciparum* reads. Human hepatitis B and primate erythroparvovirus-1 viral sequences were identified in fifteen and three mosquito specimens, respectively. 478 of the 1,142 specimens were found to contain bacterial reads and bacteriophage-related contigs were detected in 27 specimens. This analysis demonstrates the capacity of metagenomic approaches to elucidate important vector-host-pathogen interactions of epidemiological significance.

The goal of the *Anopheles gambiae* 1000 Genomes (Ag1000G) Project was to determine the genetic diversity and population structure of *An. gambiae* complex mosquitoes, the primary vectors of human malaria parasites throughout sub-Saharan Africa[1,2]. Whole genome sequencing of individual field-collected mosquitoes from thirteen African countries was used to identify the presence and distribution of single nucleotide polymorphisms conferring phenotypic traits such as reduced susceptibility to insecticides[1]. In addition to mosquito DNA, endogenous and acquired viral, bacterial, fungal, protozoan, and in cases where mosquitoes were blood-fed, vertebrate DNA sequences may also be present in data generated from these field-derived specimens.

*Anopheles gambiae* mosquitoes interact with a myriad of vertebrates, plants, and their associated microbiota at various stages of development and across a diverse range of habitats[3]. Shifts in *An. gambiae* gut microbiome composition is strongly correlated with transitions from aquatic larval habitats to terrestrial settings where adult mosquitoes are actively seeking nectar and vertebrate host blood to support flight and egg production[4]. Partially digested blood meals often contain intact and degraded RNA and/or DNA derived from the vertebrate host(s) as well as pathogens and microorganisms present in the circulatory system of the host at the time of feeding[5]. Methods such as mitochondrial DNA barcoding, amplicon-based and shotgun metagenomic sequencing, and ELISA-based approaches have been successfully used to elucidate host-feeding patterns of hematophagous insects[6]. Shotgun metagenomic sequencing offers significant advantages with respect to greater resolution and accurate identification of microbial genera and reduced primer bias as compared to amplicon sequencing

[1]Microsoft Premonition, Microsoft Research, Redmond, WA 98052, USA. [2]Department of Biological Sciences, Purdue University, West Lafayette, IN 47907, USA. [3]The W. Harry Feinstone Department of Molecular Microbiology and Immunology, Johns Hopkins Malaria Research Institute, Johns Hopkins Bloomberg School of Public Health, Baltimore, MD 21205, USA. [4]Department of Biological Sciences, University of Pittsburgh, Pittsburgh, PA 15260, USA. ✉e-mail: michael.reddy@microsoft.com

approaches, including 16S rDNA methods[7,8]. Metagenomic analyses of hematophagous insects and their blood meals represent a promising approach for rapid detection and identification of previously undescribed and established microbiota, including pathogens and their respective vertebrate hosts[9–12].

Recent advances in high throughput, deep sequencing, and bioinformatics have given rise to metagenomic approaches for rapid, highly accurate resolution of complex environmental specimens[13,14]. Microsoft Premonition has developed a Bayesian mixture model-based (BMM) metagenomics pipeline capable of identifying known taxa at the species level and estimating species present in a single specimen. The pipeline utilizes a ten-tera-base genomic reference database and cloud-scale statistical machine learning to quickly: (1) build probabilistic assignments from reads to species based on sequence similarity, (2) refine species probabilities for ambiguous reads by computing a global statistical model across all reads, and (3) identify previously undescribed, unexpected, and contaminant genetic material by aligning against all taxa with available (partial) genomic references, i.e., without a priori assumptions on which taxa might be present in a specimen and without limiting the analysis to a small subset of genomic references (e.g., to only pathogens for computational reasons)[15,16]. To this end, we analyzed the publicly available Ag1000G Phase 1 and 2 datasets using the Microsoft Premonition metagenomics BMM pipeline to determine the constituent species present in each of 1142 field-collected *Anopheles gambiae* mosquito specimens.

## Results and discussion
### Summary of BMM analysis for Ag1000G datasets
The BMM pipeline computed genome posterior probabilities (i.e., the output BMM) for over 147 billion sequence reads from 1142 individual specimens that comprise the Ag1000G Phase I and Phase 2 datasets (Table 1, Fig. S1)[17,18]. These reads were compared against a database of >600,000 reference genomes (at time of analysis) spanning the entire tree of life. The pipeline considered vertebrates, plants, protozoans, chromists, and archaea references, in addition to bacteria, viruses, and arthropods, to estimate taxon abundance profiles per mosquito[19]. An exceedingly small proportion of reads were assigned to plants, fungi and other taxa. Less than one percent of all reads (0.86%) failed assignment (with edit distance 20 or better) to any sequence present in sequence databases at the time of analysis. In the following summary, we say "a read was assigned to a taxon" to mean that a given read had the highest probability of coming from a given taxon, even though the output BMM presents possible alternatives and their corresponding probabilities.

In total, 93.02% of reads were assigned to the phylum Arthropoda, which includes *An. gambiae* mosquitoes. In addition, we detected a considerable number of reads assigned to chordates, bacteria, and bacteriophages (Table 1). Specimens associated with chordates included reads assigned to hominid, bovid, canid, equid, and phasianid hosts (Table 2). These specimens contained reads covering at least 25% of the chordate reference genomes. All specimens contained reads assigned to human sequences, however the number of reads varied widely between specimens suggesting that some of these represented blood meals taken by the mosquitoes prior to capture, while others were likely the result of specimen contamination.

### *Anopheles gambiae* species complex mosquitoes
We evaluated BMM assignments of specimens that were morphologically identified at the time of collection and genomically verified by the Ag1000G consortium to members of the *An. gambiae* species complex, which are evolutionarily similar. The *An. gambiae* species complex represented 93.3% of the probability mass given to Arthropod-assigned reads, with the remaining mass scattered around other anophelines. Next, we evaluated BMM probabilities at the species level. Higher *An. gambiae* probabilities corresponded to *An. gambiae* specimens (and similarly for *An. coluzzii* probabilities and specimens), though probabilities were more evenly distributed across these taxa, expressing greater uncertainty in the estimates

(Fig. 1). Straightforward selection of clusters from BMM statistics correctly grouped the Ag1000G specimens at the species level with 96.4% accuracy (Fig. S2). In summary, our model provided useful probabilities suggesting a robust interpretation of noisy alignment data across references with varying size, quality, and homology. Uncertainty increased at the species level which encourages more careful inspection and interpretation of model probabilities. As genomic databases such as The Darwin Tree of Life Project, VectorBase, the i5K initiative, and others expand, we expect the uncertainty of assignments to accordingly decrease[20–22].

### Vertebrate hosts
Our model assigned vertebrate reads to humans as well as several domesticated animals, including cow, goat, dog, donkey, and fowl, which all comprise common livestock species found in rural settings across Africa (Table 2)[23]. Although some human reads may be attributed to blood meals acquired from humans, it is difficult to discern human contamination from field and laboratory handling of individual specimens versus a human-derived blood meal. In contrast, the reads assigned by BMM to other vertebrates likely constitute evidence of blood feeding on non-human hosts. These signals represent over 613 million reads, sharing greater than 99% identity with vertebrate reference genomes deposited in GenBank and RefSeq databases. We suggest this indicates blood feeding by these mosquitoes had occurred on singular and, in some cases, multiple hosts. A total of 162 mosquito specimens contained non-mosquito reads corresponding to at least one chordate genome. 138 specimens contained human reads, and 24 specimens contained non-human vertebrate reads. Blood meal hosts were assigned where vertebrate reads within a mosquito specimen corresponded to at least 25% coverage of a specific reference genome sequence (Table 3). In several cases, the pipeline assigned reads to hosts based on very high coverage rates. For example, specimen ERS224451 contained 125,897,307 assigned reads covering 84% of the *Capra aegagrus* (wild goat) genome[24]. Specimen ERS224085 contained 49,266,543 assigned reads covering 84% of the *Equus asinus asinus* (donkey/ass) genome, and specimen ERS224472 contained 71,332,843 assigned reads covering over 69% of the human genome[25,26].

Given the high degree of anthropophily generally observed in *An. gambiae* and *An. coluzzii*, the abundance of non-human host signals is greater than expected (Fig. 2)[27,28]. This finding suggests that *An. gambiae* and *An. coluzzii* may be more opportunistic feeders than has previously been appreciated[27–30]. This may also reflect the relative abundance and diversity of hosts available to host-seeking mosquitoes in the sites where specimens were collected, as well as the method of collection and handling of specimens prior to sequencing. All these factors must be considered when interpreting the results and further indicate the need for accurate and extensive metadata at the time of collection[31].

### Plasmodium
The BMM pipeline assigned 5,344,273 reads (0.004%) to seven *Plasmodium* parasite species (with varying probabilities) distributed among 485 of the 1142 mosquitoes. However, the number of reads per specimen varied widely. To further examine the presence of *Plasmodium falciparum*, the most lethal and primary human malaria *Plasmodium* species transmitted by *An. gambiae*, sequence reads from each specimen were realigned using SNAP aligner against a single *P. falciparum* reference (GCA_001861075.1)[32].

### *P. falciparum* core and apicoplast genome coverage
As the *P. falciparum* nuclear genome consists of low complexity sequences (80.6% A + T), which can result in ambiguity in sequence assignments and likely was the reason for assignment to seven parasite species, we assessed the coverage of the *P. falciparum* core genome (hypervariable and subtelomeric regions excluded, 20.8 Mb; relative to the apicoplast genome, 35 kb)[33,34]. The *P. falciparum* apicoplast genome has a higher copy number compared to its nuclear counterpart (15:1 ratio). Therefore it is expected to have an increased sensitivity of detection[35,36]. Four hundred thirty-two and

148 specimens of the 1142 mosquitoes contained sequence reads mapping to the *P. falciparum* core genome and the apicoplast, respectively (Fig. 3a). As expected, all specimens with apicoplast reads had greater depth and a higher percent coverage than core genome assemblies, reflecting the disparity in sizes and copy numbers between the two genomes.

### Table 1 | Total number and percent total of reads recovered by the BMM pipeline

| Phylum | Number of reads | % of total reads |
|---|---|---|
| Arthropoda | 140,371,534,239 | 93.02% |
| Chordata | 4,713,450,496 | 3.12% |
| Bacteria | 834,882,394 | 0.55% |
| Plasmodium | 5,344,273 | 0.00% |
| Viruses | 2,040,225 | 0.00% |
| Plantae, Fungi, Other | 24,739,773 | 0.02% |
| Not clean[a] | 3,655,091,803 | 2.42% |
| Unaligned | 1,299,025,854 | 0.90% |
| **Total** | **150,900,764,783** | **100.00%** |

[a]Flagged and removed by PrinSeq or Cutadapt.

### Table 2 | Percentage of vertebrate (Phylum: Chordata) host reads identified in mosquito specimens ($n = 4,713,450,496$)

| Family (Genus) | % of total Chordata reads |
|---|---|
| Hominidae (Homo) | 82.3% |
| Bovidae (Bos, Capra) | 13.7% |
| Canidae (Canis) | 1.2% |
| Equidae (Equus) | 2.1% |
| Phasianidae (Gallus) | 0.7% |
| **Total** | **100.0%** |

## *Plasmodium* read validation

We discovered that some of these suspected *Plasmodium* reads originated from six anopheline specimens morphologically identified as male mosquitoes. Whereas only female mosquitoes feed on blood, the assignment of *P. falciparum* reads to male mosquitoes should be considered erroneous due to contamination or mislabeling. All specimens with reads aligned to *P. falciparum* apicoplast also contained reads aligned to the core genome; however, the reverse was not true. Read validation was further accomplished by establishing a threshold of the coverage for the apicoplast and core genomes in relation to the *P. falciparum* characteristic guanine–cytosine (GC) genome content.

## The positive threshold for the presence of *P. falciparum*

We determined that the GC content of *P. falciparum* consensus sequences was distinctly lower than the contigs found in the Ag1000G *An. gambiae* metagenomes and specifically when compared to bacterial taxa (Fig. 3b). Furthermore, the correlation between the GC content and the percentage of genome coverage denoted a distinct threshold in genome coverage above which the sequences had consistent GC content and within the estimated interquartile range (Fig. 3b). Specifically, for the apicoplast and the core genomes the thresholds were estimated to be 3.0% (Fig. 3c) and 0.4% (Fig. 3d), respectively. Based on these thresholds, a total of 59 specimens (5.6%) had validated coverage for both parasite genomes and were considered true positives for *P. falciparum*, while 339 specimens had no validated coverage and are likely false positives (Supplementary Data 4: "*P. falciparum* reads"). All male mosquitoes were in the latter category.

## Viruses

The BMM pipeline assigned 2,039,560 reads (0.001%) to 80 species of viruses and bacteriophage distributed among 223 of the 1142 mosquito specimens (Supplementary Note 1—"Detection of viral and bacteriophage species by the BMM and Integrator pipelines"). Eukaryotic viral sequences were found in 65 specimens, bacteriophage-related sequences were present in 167 specimens, while both eukaryotic viruses and bacteriophage were detected in 10 specimens (Supplementary Data 4: "All viral species detected" & "Viral species by specimen"). Analysis revealed that many of these detections were false positives due to physical contamination or computational misassignment. The evidence for labeling taxa as a contaminant is summarized in Column G of Supplementary Data 4: "All viral species

### Relationships between confirmed *Anopheles* species and BMM probabilities

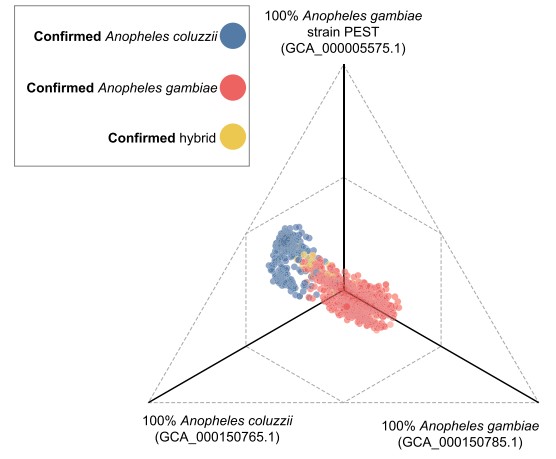
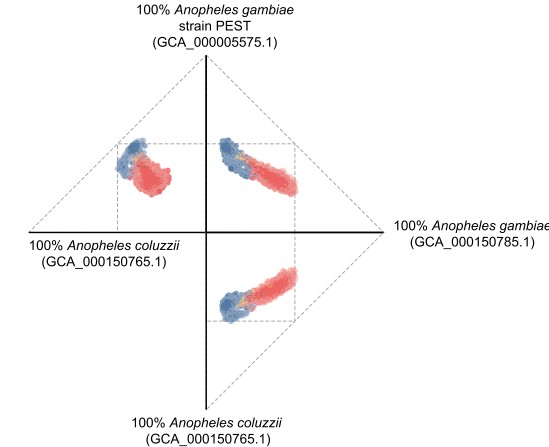

**a** Joint-distribution across 3 genomes    **b** 2-D projections

**Fig. 1 | Relationships between confirmed *Anopheles* species and BMM probabilities.** Shows how BMM distributed species probabilities across the three *An. gambiae* s.l. genomes used at time of analysis. For every read in a sample, BMM assigns some probability of that read being drawn from each of these genomes. Every 3-D data point is the sum of these probabilities across all reads in a sample, then normalized to one. **a** BMM probabilities follow confirmed species, and the probabilities assigned to confirmed hybrids generally sit in between the probabilities assigned to confirmed *An. gambiae* and *An. coluzzii*. **b** 2-D projections of this 3-D space.

detected". Many of these are well-known contaminants present in many metagenomic studies[37]. In each of these cases, a genome coverage map was generated. In many contaminant taxa, sequences map to a single genome feature that is commonly used in viral vectors. For example, we detected human cytomegalovirus (CMV) in some specimens, but all the sequence reads mapped to the immediate early promoter. SV40 was detected in some specimens with all sequence reads covering the polyadenylation signal. Similarly, we found sequences aligning to human adenovirus and avian leukosis virus, but in each case, the alignments were to vector-associated features. Close examination of viral coverage maps, and sequencing flow cell history suggested that the human immunodeficiency virus (HIV) and influenza sequences detected were the consequence of contamination, most likely stemming from previous sequencing runs using the same flow cell. No known mosquito viruses were detected in any of the specimens. However, nineteen mosquitoes contained authenticated vertebrate viral sequences, including fifteen specimens containing human hepatitis B virus (HBV), a single specimen containing ungulate erythroparvovirus-1 and three specimens containing primate erythroparvovirus-1. (Figs S3–S5; Supplementary Notes 2–4- "Presence of HBV", "Presence of ungulate erythroparvovirus-1", "Presence of primate erythroparvovirus-1"). These viruses are not known to replicate in mosquitoes; therefore, HBV reads detected were most likely present in the blood meal. In support of this notion, all fifteen HBV-positive specimens, as well as the three specimens harboring primate erythroparvovirus-1, contained human DNA while the sole specimen, ERS248730, in which ungulate erythroparvovirus-1 was detected also contained bovine DNA (Fig. S6; Supplementary Note 5—"Comparison of Premonition BMM and Kraken2 pipelines for resolving a complex specimen").

To further examine viral sequences detected by the BMM pipeline, all reads from each specimen were assembled and examined by Integrator, an extension of the Premonition pipeline for virus and microbe detection that probes amino acid similarities Supplementary Data 4: "Viral species detected-Pickaxe" & "Virus detection—Pickaxe vs BMM")[38]. Integrator confirmed the presence of HBV in twelve specimens, as well as the presence of ungulate erythroparvovirus-1 in a single specimen (Fig. 4; Supplementary Data 4: "Virus detection- Pickaxe vs BMM" & "Confirmed eukaryotic viruses") and plots were generated in Circos[39]. Furthermore, the assembled contigs contained open reading frame structures consistent with HBV and ungulate parvovirus presence. Some specimens contained near-complete genomes of HBV and ungulate erythroparvovirus-1 (Fig. 5). Integrator also uncovered numerous previously unidentified bacteriophages (Supplementary Note 6- "Novel bacteriophage"; Supplementary Data 4: "Potential novel bacteriophage"). In addition, Integrator found sequences with distant similarities to known mosquito viruses such as *Anopheles annulipes* orbivirus and Wuhan insect virus 23 (Supplementary Note 7—"Viruses with RNA genomes"; Supplementary Data 4: "Viral species detected-Pickaxe"). However, these are RNA genome viruses and may be present as integrated partial viral genomes.

## Bacteria

We analyzed bacterial taxa present in the Ag1000G Phase 1 and 2 data sets. The BMM analysis assigned approximately 0.6% of all sequence reads to bacteria (Table 1). Reads associated with bacteria were present in all 1142 specimens. However, the number of bacterial reads per specimen varied widely (Fig. 6a). Bacterial sequences may originate from microorganisms associated with the living mosquito specimens, microorganisms that grow postmortem on a preserved specimen, or due to contamination during nucleic acid preparation and sequencing. Therefore, we also examined the data following the removal of taxa commonly associated with contamination[40]. Furthermore, we only analyzed specimens with fewer than one million total bacterial reads and families with at least five thousand reads. This arbitrary cutoff was selected based on when (1) the number of bacterial reads flattened out and (2) examination of low-read specimens revealed suspected contaminants. This reduced the number of specimens containing bacterial sequences to 478 of 1142. These reads were distributed among 59 bacteria families (Fig. 6b). The presence of bacteria in Ag1000G mosquitoes was confirmed by analysis with Integrator. Bacterial contigs were only found in specimens that BMM identified as harboring bacterial reads. The bacterial phyla and genera identified by Integrator were similar to those detected by BMM (Supplementary Data 4: "All viral species detected"). As proof of concept, we examined two bacterial species, *Elizabethkingia anophelis* and *Thorsellia anophelis* in detail since both have been associated with the *Anopheles* microbiome[41,42]. *E. anophelis* was detected in 35 specimens, and *Thorsellia anophelis* in 42 of the 1142 specimens (Fig. 6c, d). Greater than 95% coverage of these bacterial genomes was achieved in a subset of the specimens. These bacteria

### Table 3 | Number of specimens containing vertebrate host reads

| Family (Genus) | # of specimens containing Chordata reads[a] |
|---|---|
| Hominidae (Homo) | 138 |
| Bovidae (Bos, Capra) | 16 |
| Canidae (Canis) | 5 |
| Equidae (Equus) | 2 |
| Phasianidae (Gallus) | 1 |
| **Total** | **162** |

[a]**≥25% coverage of a chordate reference genome**

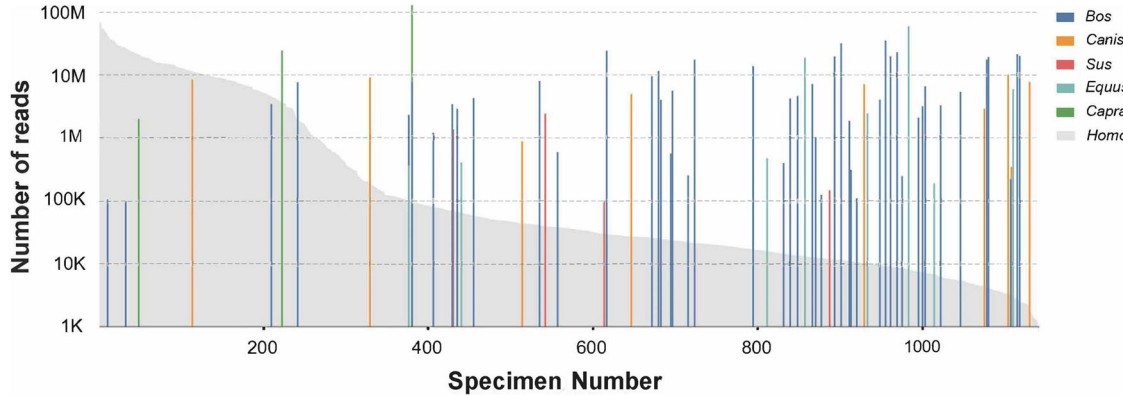

**Fig. 2 | Vertebrate host sequences.** Distribution of human and other vertebrate host sequence reads. The *y*-axis represents the number of reads assigned to the vertebrate host genus present in an individual mosquito specimen. The *x*-axis represents the individual mosquito specimens (*n* = 1142 mosquitoes). The gray area under the curve represents the number of reads assigned to *Homo sapiens* in each specimen.

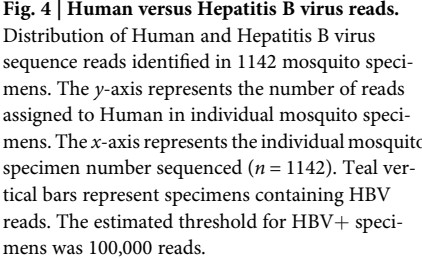

**Fig. 3 | *Plasmodium falciparum* read assignment in the Ag1000G dataset.**
**a** Distribution of percent coverage for the *P. falciparum* apicoplast and core genomes among 432 and 148 specimens of the 1142 *An. gambiae* mosquitoes. The *y*-axis represents the number of specimens (log scale) with apicoplast and core genomes reads and *x*-axis represents the percent of genomes covered ≥1×. **b** Comparison of GC content among *P. falciparum* consensus sequences and bacterial contigs found in the 1142 *An. gambiae* mosquito metagenomes. The *y*-axis represents the percentage of GC content and *x*-axis represents taxa (*P. falciparum* and bacterial families). The

bold line represents the median GC content, the box represents the interquartile range, the whiskers represent the range and dots are outliers. **c** Distribution of percent coverage as correlated to the GC content for the apicoplast genome data. The line at 3% shows the estimated threshold beyond which there is a sufficient number of reads to have distinctly *P. falciparum* GC content. **d** Distribution of percent coverage as correlated to the GC content for the *P. falciparum* core genome data. The estimated threshold is 0.4%.

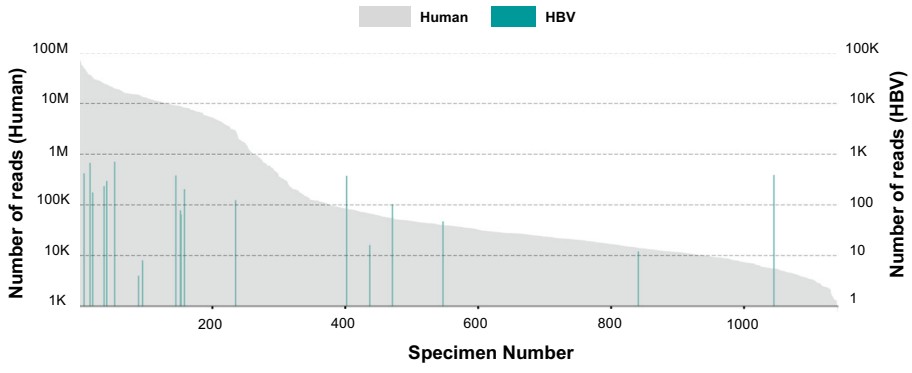

**Fig. 4 | Human versus Hepatitis B virus reads.**
Distribution of Human and Hepatitis B virus sequence reads identified in 1142 mosquito specimens. The *y*-axis represents the number of reads assigned to Human in individual mosquito specimens. The *x*-axis represents the individual mosquito specimen number sequenced (*n* = 1142). Teal vertical bars represent specimens containing HBV reads. The estimated threshold for HBV+ specimens was 100,000 reads.

accounted for most of the bacterial species detected in some specimens with no specimens having signatures of both bacterial taxa. In addition to bacteria, the BMM/Integrator analysis also identified bacteriophage contig sequences. Most bacteriophages found in nature are unidentified species distantly related to sequences in databases and therefore, most of these unique species will be missed by BMM because of alignment

requirements. This is consistent with the results of Integrator analysis that identified multiple contigs encoding bacteriophage-related proteins. Our approach does not distinguish between bacteriophage sequences present because of an ongoing infection from those that are integrated in bacterial genomes. In total, bacteriophage-related contigs were detected in 27 specimens, all of which also contained a high number

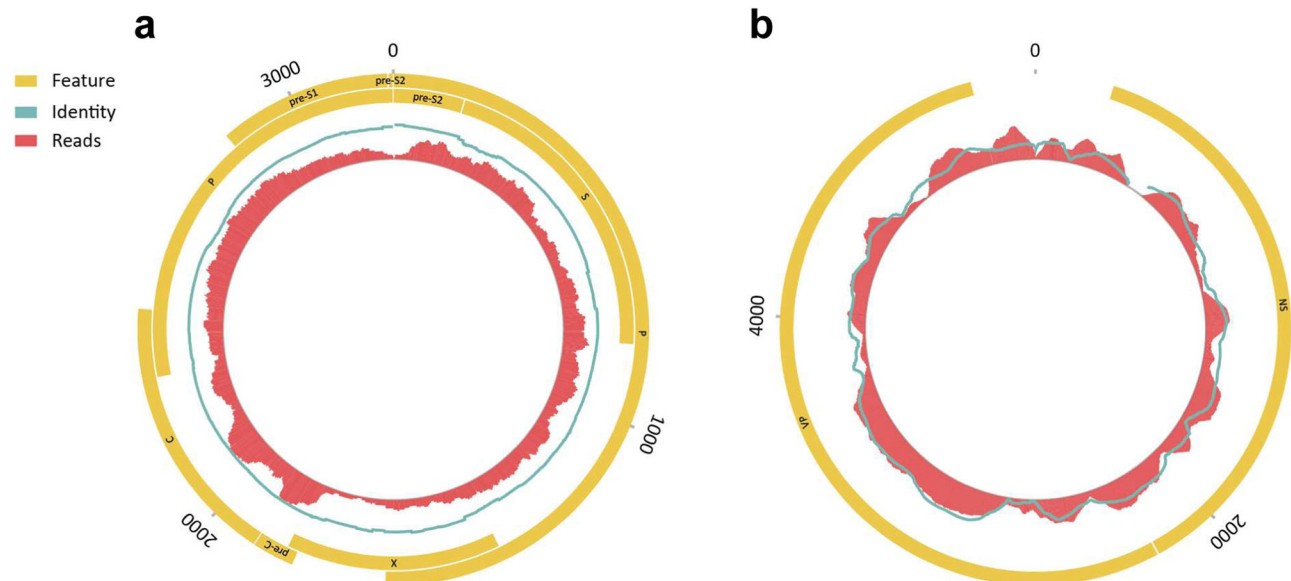

**Fig. 5 | Hepatitis B virus and ungulate erythroparvovirus-1 coverage maps. a** Hepatitis B virus. **b** Ungulate erythroparvovirus-1. Features shown in yellow represent open reading frames encoding viral proteins. The teal line represents the percent identity to the reference genome. Red blocks represent the number of sequence reads.

of bacterial reads (Fig. 6a). In three of the 27 specimens, Integrator detected and assembled bacteriophage contigs despite not having any BMM-assigned bacteriophage reads.

## Conclusions

As projects like Ag1000G continue to expand the volume of genomic data available for the *An. gambiae* species complex across its range of distribution, we expect our mixture model to become increasingly accurate at resolving species-level assignments. Our Bayesian mixture model assigned vertebrate reads to humans as well as several domesticated animals, including goat, cow, dog, and donkey hosts. In addition, evidence of mixed blood meals derived from two host species was detected in several specimens. We were able to ascertain reads assigned to *P. falciparum* in several vector specimens. The relatively low coverage of the 23 Mb *Plasmodium* parasite genome demonstrated the challenges of detecting a small fragment of the parasite in a large specimen of the host genome and mixed DNA templates. However, to the best of our knowledge, using the detection of both *Plasmodium* core and apicoplast genomes proved to be an original method to validate parasite presence. Yet, because whole mosquitoes were typically used for DNA extractions and sequencing, it is not possible to discriminate whether mosquitoes were infectious.

We were unable to ascertain whether the wide-spread presence of reads assigned to *Homo sapiens* in nearly all specimens was a consequence of human feeding or contamination during field collection, sorting and identification, laboratory manipulation, and nucleic acid extraction or due to residual contamination of NGS flow cells between sequencing runs. Specimens originated from many different collectors and were handled and extracted using multiple approaches. Some specimens were stored in ethanol, while others were desiccated; some were extracted soon after collection, and others after extended storage. Thus, bacteria may be present as part of the mosquito microbiome, phoretic on the external surface of the insects, as contaminants introduced during collection/specimen processing, or even microbial growth during specimen storage. Thus, we term these as "mosquito-associated bacteria". The enormous number of bacterial reads present in some specimens suggests that bacteria were actively growing in some specimens.

The microbiome of individual mosquitoes is relatively simple from this data set. Since bacteriophage cannot grow in insect cells, bacteriophage sequences should only be present in specimens containing bacteria.

However, we cannot distinguish bacteriophage infection from integrated phage genomes. All the bacteriophages detected were previously undescribed. The high degree of sensitivity of the NGS method underscores the need to preserve specimen integrity and standardize approaches from collection through analysis to accurately determine sequence identity and the nature of biological associations. Additionally, the results demonstrate the importance of targeting collections and metadata to address specific questions. We continue to investigate unusual genomic assignments for systematic contamination of reference databases and are developing disciplined methods to address reference contamination. This study shows that metagenomic analysis of mosquitoes provides a robust strategy for detecting and monitoring the host species from which mosquitoes obtain a blood meal, as well as protozoa, bacteria, and viruses that are circulating among vertebrate hosts.

## Methods
### Statistics and reproducibility

Briefly, the Microsoft Premonition BMM pipeline takes as input: (1) a sample $X = \{x_1, \ldots, x_m\}$, which is a collection of sequencer reads, and (2) a reference genome database $Ref = \{g_1, \ldots, g_n\}$, which is a set of genomes. It computes the probability distribution $p(r, g|X, Ref)$, which is the probability that read $x$ in the sample $X$ came from genome $g$ in the reference database $Ref$. This distribution is computed without assumptions on the species that may be present in the sample (and so every read is aligned to every genome in the reference). This is well-suited for environmental specimens that have few biological constraints on the species that might be in a specimen and that may contain genomic fragments from many species with low genome coverage. The uncertainty in the resulting probability distribution can indicate: (1) uncertainty of species due to sequence similarity, (2) the presence of undetected species where reads are unlikely to have come from any genomes in the available references, and (3) genome coverage patterns that are consistent with non-biological artifacts—as well as other phenomena. The resulting probability distribution is a Bayesian mixture model (BMM), which is described in detail in the Methods under 'Generative Model' and shown figuratively in Fig. S1. In the context of the Ag1000G dataset, this allows the pipeline to suppress the probability that low complexity or highly conserved anopheline reads might have come from other anopheline species based on the overwhelming evidence for *An. gambiae* s.s. or *An. coluzzii* coming from other unambiguous reads in the sample. At the same time, some

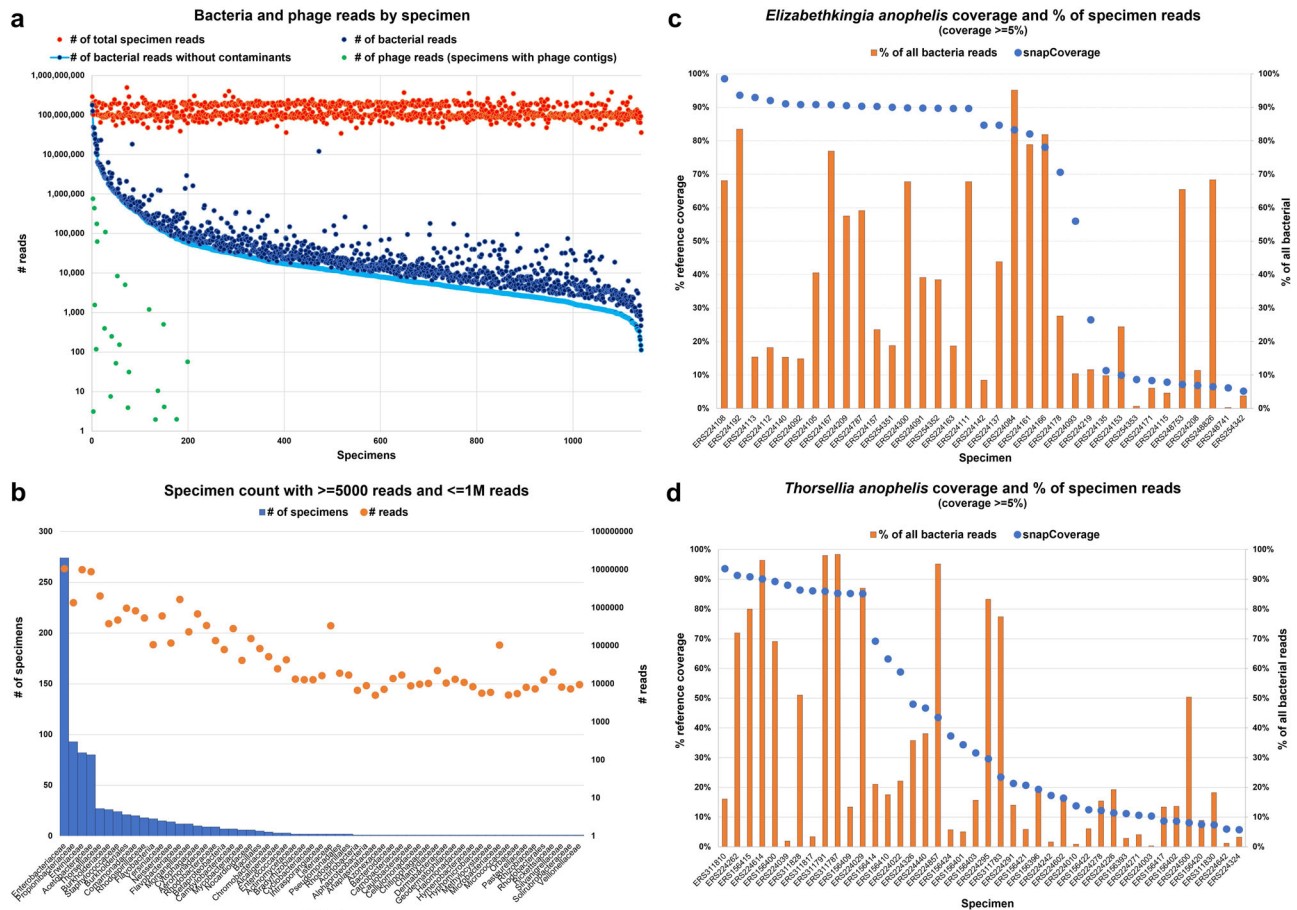

**Fig. 6 | Bacteria and bacteriophage content. a** Bacteria and bacteriophage reads found in each specimen. The *x*-axis represents the individual mosquito specimen (*n* = 1142 mosquitoes). The *y*-axis represents the number of reads as assigned by BMM. Orange—all read in the specimen, dark blue—number of bacterial, light blue —number of reads after removing common contaminants, green—number of bacteriophage reads. **b** Number of specimens and reads with bacteria. Specimens with more than 1,000,000 total bacterial reads and families with less than 5000 reads within the specimen were excluded. The *x*-axis represents the bacteria family. The

right-*y*-axis and blue bars represent the number of specimens. The left *y*-logarithmic axis and orange dots represent the total number of reads for these specimens. **c** Thirty-five specimens with at least 5% coverage of the *Elizabethkingia anopheles* reference genome. Orange bars represent the percentage of *Elizabethkingia anopheles* reads as a fraction of all bacteria reads within an individual specimen (common contaminants excluded). Blue dots represent the percentage of coverage of the reference genome. **d** Same as (**c**) but for 42 specimens with at least 5% coverage of the *Thorsellia anopheles* reference genome.

---

small probability can be assigned to these less likely interpretations, so they are available for consideration. Various quantities can be derived from this statistical model. For simplicity, we will consider only a few of these quantities here:

- For every read $x$, the genome posterior probability $p(g|x, X, Ref)$ gives the probability that genome $g$ contributed read $x$ to the sample. These probabilities sum to one for each read, so they can also be treated as fractionally mapping reads to genomes. For instance, a read $x$ may have $p(An.gambiae|x, X, Ref) = 0.7$ and $p(An.coluzzii|x, X, Ref) = 0.3$, with all other genomes having zero probability, indicating $x$ was more likely to have come from the *An. gambiae* genome than the *An. coluzzii* genome under modeling assumptions.

- The expected number of genome reads is the expected number of reads a genome $g$ contributed to a sample and is the sum of all fractional reads mapped to that genome, i.e., $E_{reads}[g] = \sum_{x \in X} p(g|x, X, Ref)$. This can be extended to the expected number of reads contributed by an arbitrary taxon. Given a taxon $tx$ let $g(tx) = \{g_{tx1}, \ldots, g_{txk}\} \subseteq Ref$ be the set of all genomes in the reference database that belong to that taxon. Then, the expected number of reads for that taxon is $E_{reads}[tx] = \sum_{g \in g(tx)} E_{reads}[g]$. For example, $E_{reads}[diptera] - E_{reads}[Anopheles]$ gives the expected number of reads contributed from non-anopheline dipterans.

- We extend the standard definitions of genome percent identity (i.e., the average percent identity of all reads assigned to a genome) and of genome coverage (i.e., the total fraction of genome locations for which at least one read is assigned to that location) to the BMM setting, where reads are not assigned with 100% probability.

- The $\epsilon$-genome coverage and $\epsilon$-genome percent identity of a genome $g$ are the genome coverage and genome percent nucleotide identity calculated using the set of all reads with genome posterior probability greater than or equal to $\epsilon$, i.e., $\{x \in X | \, p(g|x, X, Ref) \geq \epsilon\}$. Choosing a smaller value for $\epsilon$ yields a higher coverage because more reads are considered, but a lower value for percent identity because more divergent reads are included. Posterior credible intervals could also be defined. For this presentation "x% of reads were placed in taxon" means the percentage of expected reads contributed by that taxon to the total number of reads in that sample, i.e., $100 \times \frac{E_{reads}[tx]}{|X|}$.

### Generative model

**Genomes and mixture probabilities.** Let $g = (g_1, \ldots, g_m)$ be a vector of reference genomes for $m$ distinct species, and $w = (w_1, \ldots, w_m)$ be a vector of probabilities. Each $w_i$ is the probability of $g_i$ contributing a read to a collection of reads $X$ (i.e., a multiset of reads). The mixture

probabilities satisfy ($\sum_{i=1}^m w_i$) = 1. More precisely, each genome has length $\lambda_i$ and is a function from nucleotide positions to nucleotide characters, i.e., $g_i : \{1, \ldots, \lambda_i\} \to \{A, T, C, G\}$ where $g(j)$ is the nucleotide at position $j$. We write $\Omega$ for the alphabet of nucleotide characters.

**Partitions and partition probabilities.** To model (un-)even sampling of reads across genomes, each genome is partitioned into $\boldsymbol{p_i} = (p_{i,1}, \ldots, p_{i,n})$ parts, where each part $p_{i,j} \subseteq \{1, \ldots, \lambda_i\}$ is a non-empty disjoint interval of nucleotide positions. The union of all parts equals $\{1, \ldots, \lambda_i\}$. Let $\boldsymbol{\pi_i} = (\pi_{i,1}, \ldots, \pi_{i,n})$ be a vector of probabilities, where $\pi_{i,j}$ is the probability that part $p_{i,j}$ contributes a read to a collection, given that this read was contributed by $g_i$. Partition probabilities satisfy that $\forall i.(\sum_{j=1}^{|\boldsymbol{p_i}|} \pi_{i,j}) = 1$.

**Reads and read probabilities.** A read $\boldsymbol{x}$ with length $l$ drawn from genome $g_i$ and partition $p_{i,j}$ starting at position $k \in p_{i,j}$, is a vector $\boldsymbol{x} = (c_1, \ldots, c_l)$ where each $c_q \in \Omega$ is the $q^{th}$ character of read $\boldsymbol{x}$. The likelihood of each $c_q$ is simply:

$$\Pr\left(c_q = y \middle| \begin{array}{l} genome = g_i, \\ part = p_{i,j} \\ pos = k, \\ mismatch = \gamma_i^{miss} \end{array}\right) = \begin{cases} 1 - \gamma_i^{miss}, & y = g_i(k+q-1) \\ \frac{\gamma_i^{miss}}{|\Omega|-1}, & otherwise \end{cases}$$

where $\gamma_i^{\text{miss}}$ is the probability that any sampled nucleotide $c_q$ disagrees with the reference. We do not treat $\gamma_i^{miss}$ as a raw sequencer error rate, but as an overall parameter that encompasses many possible sources of reference mismatches. For brevity, we will not formalize insertions and deletions here.

**Algorithm: sampling the mixture.** Given a specification of the above parameters, then the mixture is sampled as follows.
1. Draw a genome according to mixture weights: $z_g \sim \text{Mult}(1; \boldsymbol{w})$.
2. Draw a partition according to partition probabilities of that genome: $z_p | z_g \sim \text{Mult}(1; \boldsymbol{\pi}_{z_g})$.
3. Uniformly draw a position $k$ from the chosen partition: $z_k | z_g, z_p \sim \text{Uniform}(z_p)$.
4. Draw a read $\boldsymbol{x}$ of length $l$, one element at a time: $c_q | z_g, z_p, z_k \sim \text{Mult}(1; p_A, p_T, p_C, p_G)$.

In the above, $\text{Mult}(n; \boldsymbol{q})$ denotes the multinomial distribution with $k$ classes, $n$ trials, and class probabilities $\boldsymbol{q} = (q_1, \ldots, q_k))$. $\text{Mult}(1; \boldsymbol{q})$ is the categorical distribution, which we treat as drawing a single class. Finally, in step 4, $p_A = \Pr(x_q = A | z_g, z_p, z_k, \gamma^{miss})$, $p_T = \Pr(x_q = T | z_g, z_p, z_k, \gamma_g^{miss})$ and so on, for the remaining nucleotides.

**Model fitting.** Given an observed vectors of reads $\boldsymbol{X} = (\boldsymbol{x}_1, \ldots, \boldsymbol{x}_n)$, the goal of the BMM algorithm is to construct a maximum likelihood estimate (MLE) of the genome mixture probabilities $\boldsymbol{w}$ that maximize $\Pr(\boldsymbol{X} | \boldsymbol{w}\Phi)$, where $\Phi$ are the remaining model parameters that are treated as fixed values not subject to estimation (e.g., partition probabilities and mismatch probabilities).

**Expectation–maximization (EM) algorithm.** Computing the MLE is non-trivial because the observed data are incomplete—they do not include the source genomes, partitions, and starting positions of the observed reads (i.e., the variables $z_g, z_p, z_k$ in the generative model are not observed). The approach we utilize is the EM algorithm, which alternates between estimating the unobserved variables given an estimate of the $\boldsymbol{w}$, and then adjusts $\boldsymbol{w}$ to better fit the observed data—until convergence. The details of this algorithm, and its formal guarantees, are outside the scope of this description. We provide the key ingredients required by the EM algorithm.

**Joint distribution of observed and latent variables.** To estimate these parameters, EM requires a joint distribution of the observed data (i.e., the

reads $\boldsymbol{X}$) and the unobserved data (i.e., latent variables $z_g, z_p, z_k$) given model parameters. Let $\boldsymbol{Z}$ be a zero-one matrix with dimensions $|\boldsymbol{g}| \times (\max_i \lambda_i) \times |\boldsymbol{X}|$. The entry $Z_{i,j,k} = 1$ if genome $g_i$ at position $j$ contributed read $x_k$, otherwise it is zero. The joint distribution derives from the generative model as follows:

$$Pr(\boldsymbol{X}, \boldsymbol{Z} | \boldsymbol{w}; \Phi) = \prod_{i=1}^{|\boldsymbol{g}|} \prod_{j=1}^{\max_i \lambda_i} \prod_{k=1}^{|\boldsymbol{X}|} \left[ w_i \cdot \frac{\pi_{i,part(i,j)}}{\left|p_{i,part(i,j)}\right|} \cdot Pr\left(\boldsymbol{x}_k, |, g_i, p_{i,part(i,j)}, j, \gamma_i^{miss}\right) \right]^{Z_{i,j,k}}$$

where $part(i, j)$ returns the index of the partition containing position $j$ in genome $g_i$. From the generative model above, the probability of $\boldsymbol{x}_k$ having been drawn from $g_i$ and location $j$ is a function of its edit distance $d$ and length $l$ as follows: $\Pr(d, l) = (\gamma_i^{\text{miss}})^d \cdot (1 - \gamma_i^{\text{miss}})^{l-d}$.

**All-by-all read probabilities computation.** A non-trivial step performed by the BMM is the all-by-all computation that evaluates the probability of every read $\boldsymbol{x}_k$ coming from every location $j$ in every reference genome $g_i$. This probability matrix can be quite large and is computed once at the beginning of the algorithm. The precise number of locations stored in this matrix depends on the choice of aligner and its settings, which are parameters of BMM.

**Unbiased sequencing assumptions.** In an unbiased metagenomics setting, we expect reads to be represented from across the genomes of present taxa. This corresponds to setting a genome's partitions to be of equal weights and sizes. This simplifies the joint distribution to:

$$Pr(\boldsymbol{X}, \boldsymbol{Z} | \boldsymbol{w}; \Phi) = \prod_{i=1}^{|\boldsymbol{g}|} \prod_{j=1}^{\max_i \lambda_i} \prod_{k=1}^{|\boldsymbol{X}|} \left[ w_i \cdot \frac{1}{\lambda_i} \cdot Pr\left(\boldsymbol{x}_k, |, g_i, p_{i,part(i,j)}, j, \gamma_i^{miss}\right) \right]^{Z_{i,j,k}}$$

An important impact of this setting is that shorter genomes are preferred over longer genomes, even if a subset of reads align equally well to both. For example, this setting allows BMM to suppress (microbial or viral) contaminants in eukaryotic reference genomes, opting instead to amplify the probabilities that microbial or viral species (with shorter genomes) were truly present in the mixture.

**Practical corrections to unbiased sequencing assumptions.** In practice, even this assumption does not fully handle interactions between contamination, partial genome references, and low complexity regions to name a few challenges. In this presentation, the solution is to heuristically define an effective genome size $\lambda_i^{\text{effective}} = f_{\text{effective}}(g_i, \boldsymbol{X}, \Phi)$ according to a function that can observe the reference genome, input data, and other model parameters. The function $f_{\text{effective}}()$ can be designed using many approaches. We build ours using intuitive definitions and testing on many simulated and actual metagenomic datasets[43].

**Partitioned read dispersion.** Given the all-by-all read probabilities, the partitioned read dispersion for genome $g_i$ is the uniformity of read probabilities across its partitions. It has a value of 1 if read probabilities are equally dispersed across partitions, and a value of 0 if they are all concentrated into one partition. First, the read probability mass in a partition $j$ of genome $g_i$ is:

$$mass_{i,j} = \sum_{k \in p_{i,j}} \sum_{l=1}^{|\boldsymbol{X}|} Pr\left(\boldsymbol{x}_l, |, g_i, p_{i,j}, k, \gamma_i^{miss}\right)$$

and the partitioned read dispersion is one minus the Gini coefficient of these masses:

$$PRD_i = 1 - \frac{\sum_{j=1}^{|\boldsymbol{p_i}|} \sum_{k=1}^{|\boldsymbol{p_i}|} |mass_{i,j} - mass_{i,k}|}{2 \cdot |\boldsymbol{p_i}|^2 \cdot \widehat{mass_i}}$$

Through sweeps of functional forms on simulated and actual metagenomic data, we utilize the below correction for genome size. The exponent $p_{eff}$ brings genomes of different sizes closer together decreasing length penalties. But low PRD increases penalties by increasing the effective genome length:

$$\lambda_i^{effective} = \frac{\lambda_i^{\frac{1}{p_{eff}}}}{PRD_i^{p_{eff}}}$$

**Major settings**. Based on this discussion, the major settings for BMM were:

1. All reference genomes are initially equally likely (uniform priors).
2. Unbiased sequencing settings are assumed, i.e., all partitions were given equal weights. The effective genome lengths $\lambda_i^{effective}$ were computed with the exponent $p_{eff} = 4$ based on sweeps of simulated and actual metagenomic data.
3. The per-read miss probability was set to $\gamma_i^{miss} = 0.1$ for all genomes based on sweeps of simulated and actual metagenomic data. Of consideration was ensuring more distant taxa could receive probability mass, given overall sparsity of genome references.

### Metagenomic analyses

The Microsoft Premonition BMM pipeline was applied to DNA sequencing datasets of 1142 mosquitoes from the Ag1000G Phase 1 and 2 datasets[17,18] All Phase 1 and Phase 2 reads were processed as follows. First, all reads were deduplicated looking for exact and exact reverse complement duplication. The duplicity count for each read was recorded. The adapters were trimmed with Cutadapt v1.13[44]. Reads with low-quality or low complexity were removed with PrinSeq v0.20.4[45]. To reduce computational complexity reads that aligned to mosquito references with an edit distance of five or better were subsampled at a rate of 1.0%. The references for subsampling consisted of *An. gambiae* (g4 assembly; GCA000150785.1), *An. coluzzii* (m5 assembly; GCA000150765.1) and *An. gambiae* str. PEST (AgamP3 assembly; GCF000005575.2)[46,47]. All reads were aligned with SNAP-aligner with an edit distance limit of up to 20 against the selection of RefSeq and GenBank assemblies (615,026 total accessions retrieved June 2018)[32]. The selection aimed to have at least one high-quality assembly for every species taxonomic identifier. All viral references also retrieved from NCBI's GenBank in June 2018 were included. A metagenomic assignment of reads to accessions was computed based on a BMM implemented as an EM algorithm and extended with a heuristic that prefers accessions with uniform coverage. An accession was assigned to each read producing a probabilistic BMM call, revealing the most likely taxonomic assignment.

### Integrator pipeline

For all Ag1000G specimens, we applied these steps: (1) All reads classified by BMM as bacterial or unaligned were assembled with SpaDES v3.14[48]. Contigs of a minimum length of 2000 bp were analyzed, producing probable bacterial contigs. (2) Probable bacterial contigs were aligned with Diamond v0.9.24.125 aligner against the RefSeq non-redundant (nr) protein database[49]. All Diamond matches for a given contig were aggregated at the desired taxonomic level. (3) The taxon with the highest integral of percent identity over contig length was assigned to each contig, resulting in an Integrator assignment for a probable bacterial contig. Steps one through three were reiterated for a set of BMM viral and unaligned reads and produced an Integrator assignment for each probable viral or bacteriophage contig.

### Reporting summary

Further information on research design is available in the Nature Portfolio Reporting Summary linked to this article.

### Data availability

The datasets analyzed in the current study are available via the *Anopheles gambiae* 1000 Genomes Consortium website: Ag1000G phase 1 AR3.1 data release: https://www.malariagen.net/data/ag1000gphase1-ar3.1. Ag1000G phase 2 AR1 data release: http://www.malariagen.net/data/ag1000gphase2-ar1 Source data for figures and tables are included in this published article and its Supplementary Data files 1–4. All other data are available from the corresponding author on reasonable request.

### Code availability

The Microsoft Premonition pipeline is a proprietary cloud service, and APIs to access this service are made available to select partners through the Microsoft Premonition Early Access Program (terms and conditions apply, see http://microsoft.com/premonition for details). If needed to assist reviewers, authors will provide the computed mixture model statistics, aggregated at the specimen level as a data artifact upon request to the corresponding author. Access to read-level data is managed by the Ag1000 Genome Consortium (terms and conditions apply) and should be requested directly from the Ag1000 Genome Consortium.

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

## Acknowledgements

The authors gratefully acknowledge the Ag1000G Consortium for making the Phase 1 and 2 datasets publicly available. We are indebted to consortium members Mara Lawniczak and Alistair Miles for their valuable comments and guidance. We also thank Christian Gauthier, Renee Ali, and Simon Frost for their insightful comments. Authors A.P., X.C., M.R.R., I.H., and E.K.J. are paid employees of Microsoft Corporation. The funder provided support in the form of salaries for authors [I.H., A.P., X.C., M.R.R., and E.K.J.], but did not have any role in the study design, data collection and analysis, decision to publish, or preparation of the paper. J.D. and G.C. were supported by funding from the Department of Biological Sciences at Purdue University. D.E.N. and M.E.G. were supported in part by the Johns Hopkins Malaria Research Institute and Bloomberg Philanthropies. Commercial funder Microsoft Research provided financial support to J.M.P., M.E.G., and D.E.N.

## Author contributions

A.P.: Analyzed data relating to *Anopheles gambiae* mosquitoes and vertebrate hosts, contributed to the overall structure of the study and co-author of initial draft. MRR: Analyzed data relating to *Anopheles gambiae* mosquitoes and vertebrate hosts and contributed to the overall structure of the study and co-author of the initial draft. Corresponding author. X.C.: Contributed to the generation of figures and co-author of the initial draft. I.H.: Contributed to the overall structure of the study and co-author of the initial draft. J.D.: Analyzed data relating to *Plasmodium* and generation of figures. M.E.G.: Contributed to interpretation relating to *Anopheles gambiae* mosquitoes and vertebrate hosts. G.C.: Analyzed data relating to *Plasmodium* and generation of figures. Contributed to the overall structure of the study and co-author of the initial draft. DEN: Analyzed data relating to *Anopheles gambiae* mosquitoes and vertebrate hosts, Contributed to the overall structure of the study and co-author of the initial draft. J.M.P.: Analyzed data relating to viruses, bacteriophages, and bacteria. Contributed to the overall structure of the study and co-author of the initial draft. E.K.J.: Contributed to the overall structure of the study and co-author of the initial draft.

## Competing interests

Authors A.P., X.C., M.R.R., I.H., and E.K.J. are current, salaried employees of Microsoft Corporation. This does not alter our adherence to

Communications Biology's policies on sharing data and materials. The remaining authors declare no competing interests.
