## [Peer Review File · Communications Biology]

Reviewers' comments:

Reviewer #1 (Remarks to the Author):

Reviewer comments

Summary

The work introduces a bespoke Bayesian Mixture Model (Microsoft Premonition) for taxonomic classification of metagenomics sequence data. It is used to infer the posterior probability distributions of reads in a sample that belong to reference genomes, and by extension the set of genomes for specific taxa. Thus, the model assigns sample composition to a simplex of taxa identity.

The approach is applied to a dataset of mosquitos including their blood meals as samples that represent a mixture of DNA from different hosts in addition to the particular mosquito species. Model taxa assignments were validated by the confirmed presence of known mosquito species. Other model assignments provide insight into expected and novel biological diversity of unconstrained mosquito borne environmental DNA, as well as uncertainty inherent in the blood meal community or due to sampling contamination.

Overall comments

1. Overall the work is concise, mostly clear, and organized (especially the Introduction)
2. There are no citations for the Microsoft Premonition pipeline as a method anywhere that I can see in the manuscript. Understanding that it is proprietary, I cannot tell if its specific methodology has been peer-reviewed at all. Is there a separate manuscript or reference for its methodology available, or that you can provide? Is it being introduced in this work? A search of the Microsoft website and literature did not yield any deeper technical details than what was included in the short methods section. I appreciate this succinctness but more details are needed. For example, what priors were used in the BMM? If data quality is poor (contamination) even so-called uninformative priors could have major effects on the posterior. If the data collection, storage, processing approaches introduced many sources of bias and/or error as stated in the conclusion, then the data would be very noisy so the priors and model assumptions are important to state explicitly.
3. Was a multivariate normal distribution model assumed for the data?
4. Is the method completely unsupervised?
5. Were the metagenomic reads only assigned to taxa or were any functional distributions inferred (genes, gene families, pathways)?
6. Was taxonomic assignment benchmarked by comparison to any other conventionally used pipeline (e.g., Kraken2+Bracken, MetaPhlAn (newest version), BLAST)
7. Are there added benefits of this model approach over other classification methods?
8. Is the focus of this work to highlight the Premonition BMM as employed with an interesting real-world dataset? Or, are biological insights into the Ag1000G genomes the main result?

Specific comments

9. 60-62: Is there an implication here that the method used was able to determine pathogen-host connections? The blood meal is a mixture of DNA so how is a pathogen inferred to come from a particular unknown host?
10. 81: What is meant by efficiently? Is there a relevant comparison to remark?
11. 90, 326-329: Can borderline cases, where posterior probabilities for two different taxa assignments were similar (e.g., 0.49 and 0.51), be quantified? For example, how many samples had borderline percentages and were not confirmed hybrids (middle ranges of points in Fig 1 cluster). It's hard to tell how many samples had borderline assignments but were not hybrid, which indicates uncertainty in the model.
12. 87-90, 102-104: Can this information be combined?
13. 111: Unclear which taxonomic level is being specified if not species (identified in the section title and next sentence?)
14. 106-115: Is this saying the method was validated by apriori verified sample identities compared to the model assignment probabilities? If so, how?
15. 131-135 Results and Discussion section: These statements are a bit confusing about the difference between the way '99% identity' and '25% coverage' terms are being used for these 'other' vertebrate associated samples. Please clarify. Is the 99% identity referring to the BMM (MS Premonition) classification or an independent pipeline classifier for taxa assignment? What 132 genomes are being referred to? I am guessing that 99% identity means percent of reads assigned, whereas coverage is used for specific taxon genomes.
- 339-347 Methods section: Please define percent genome identity versus coverage separately. Both are referred to, but it is only stated they are calculated using the set of all reads with genome posterior probability greater than or equal to epsilon. Can the coverage vs identity difference be explained with an example for a specific taxon?
16. There are multiple results in the Plasmodium results section. Can these be organized into separate paragraphs that start with the main point? Although I agree with a combined Results and Discussion for this work there is a lot of 'jumping around' from point to point within the section that reads in a jumbled way.
17. 301: Please show how these posterior probabilities are being calculated as an explicit expression to obtain the distribution over samples of taxa assignment.
18. 304-306: This is an interesting point and could potentially be generalized to determine environmental constraints (e.g., abiotic filtering, niche partitioning) affecting species distributions.
19. 312-313: Please provide a reference to the full description of the pipeline.
20. 314: Was this done using Gibbs sampling?
21. 313-315: This is vague. "Based on statistical evidence" What statistic? Why? How does the pipeline do this?
22. 317: Missing word "might have come [from] other anopheline species"
23. 329: Can the modelling assumptions be explicitly stated somewhere
24. 346: Number 'of' reads vice "number or reads"?

Reviewer #2 (Remarks to the Author):

Dr. Pastusiak and colleagues used a Microsoft Premonition Bayesian mixed model based (BMM) metagenomics pipeline to analyze the publicly available Ag1000G data and reported a number of interesting findings regarding species identification, hosts/Plasmodium/viruses/bacteria identification, and possible contaminations. I have a few major concerns that need to be addressed before the manuscript can be considered for publication.

- 1) The description of the pipeline is not sufficient to help ensure reproducibility or rigor. For example, I could not find what alignment program was used in the pipeline.
- 2) It is also not clear how the implementation of the pipeline significantly improved the efficiency or accuracy of the metagenomics analysis.
- 3) The authors should further explore the causes of the uncertainty/inaccuracy associated with species assignment (lines 105-122). I don't fully understand Figure 1.
- 4) How many individuals were identified as males according to sample morphology? I suggest that the authors perform molecular identification of mosquito sex using known Y-linked markers (Hall et al., 2016, PNAS). Males could serve as a good control for vertebrate contamination as they do not feed on blood. It may also be interesting to compare/contrast the microbiome of the two sexes.
- 5) Lines 193-194, please provide support for the statement.

Reviewer #3 (Remarks to the Author):

The authors leverage data from the Anopheles 1000 genome project to develop a new metagenomics Bayesian mixed model within the Microsoft Premonition pipeline. Using this approach, they make a few key findings. Specifically, that Anopheles seem to be feeding at higher rates on non-human animals than previously thought with ~25% of feeds deriving from domestic and peri-domestic animals. Another interesting finding to emerge from this dataset was the surprisingly low amounts of fungal reads associated with the samples suggesting that fungus do not make up a substantial part of the mosquito microbiome. The finding of vertebrate viruses in the samples is really cool and supports earlier work using xenosurveillance. However, this is a large analysis with no experimental validation of the pipeline.

- 1) Without a more detailed explanation of the Microsoft Premonition pipeline, the authors need to go above and beyond to demonstrate the validity of their model. This can be achieved a number of ways including performing some experiments in the lab with mosquitoes known and unknown content.
- 2) I find it curious that bacteria were not found in all of the samples whether it be the actual microbiome or contaminating bacteria considering these were collected in the field.
- 3) Also, 478 samples had bacterial reads and 485 had plasmodium reads. It's curious that both, presumably unlinked habitats, would be found in roughly the same number of samples. Is this an artifact of the quality of sequencing data from those samples compared to the rest? Without sequence quality assessments and metadata, the data is purely observational and does not actually tell us much about the metagenome of mosquitoes. Consequently, this manuscript is principally about a pipeline; a

pipeline that is protected by a proprietary cloud service.

Reviewer #1 (Remarks to the Author):

Summary: The work introduces a bespoke Bayesian Mixture Model (Microsoft Premonition) for taxonomic classification of metagenomics sequence data. It is used to infer the posterior probability distributions of reads in a sample that belong to reference genomes, and by extension the set of genomes for specific taxa. Thus, the model assigns sample composition to a simplex of taxa identity. The approach is applied to a dataset of mosquitos including their blood meals as samples that represent a mixture of DNA from different hosts in addition to the particular mosquito species. Model taxa assignments were validated by the confirmed presence of known mosquito species. Other model assignments provide insight into expected and novel biological diversity of unconstrained mosquito borne environmental DNA, as well as uncertainty inherent in the blood meal community or due to sampling contamination.

Authors' response: We thank the reviewer for their thorough review of the manuscript and succinct summary of the experimental work presented.

Reviewer 1 Overall comments (in italics):

1. Overall the work is concise, mostly clear, and organized (especially the Introduction)

Authors' Response: We thank the reviewer for their comment and appreciate the feedback provided.

2. There are no citations for the Microsoft Premonition pipeline as a method anywhere that I can see in the manuscript. Understanding that it is proprietary, I cannot tell if its specific methodology has been peer-reviewed at all.

Authors' Response: This manuscript provides the first description of the Premonition Bayesian Mixture Model (BMM) metagenomic pipeline in a manuscript submitted for publication.

3. Is there a separate manuscript or reference for its methodology available, or that you can provide?

Authors' response: The BMM pipeline has been used by another academic group to perform comparative metagenomic analyses of endogenous bat viruses in a novel induced pluripotent stem cell model as reported in Dejosez et al. Cell. 2023 Mar 2;186(5):957-74. Kraken2 and Premonition BMM pipeline analyses are visualized in supplementary figure S5; below).

Figure S5. Virome mining approaches, related to Figure 7 and Tables S5 and S6

RNA-seq reads were first classified using Kraken2 (left). The classified reads were then either analyzed directly or separated by their homology to a mammalian or a non-mammalian virus using virion. In a 'bottom-up' approach (red boxes), the reads that were classified to belong to the same virus were assembled *de novo* and blasted against databases containing all viruses (AVDB), mammalian viruses (MVDB) or non-mammalian viruses (NMVDB). The identified sequences with viral hits were then further mapped to the *R. ferrumequinum* (RhiFer) genome (gVH) and in a final step, reads/assemblies that did map to the genome were extended by 5 kb on both sides and the sequences again blasted against the databases as before to extend the search for viral hits in flanking genomic regions (fgVH). In a parallel 'top-down' approach (blue boxes), the Kraken2 reads were mapped directly to the *R. ferrumequinum* genome, micro-assemblies were generated based on the mapped reads, and blasted directly against the different databases as before or first extended by 5 kb on both sides. In an orthogonal approach, the reads were similarly assigned to viruses using the Microsoft Premonition metagenomics pipeline (right). First all reads were classified as aligning to a virus or as being unaligned, micro-assemblies were generated and mapped to viral genomes (red box). Consensus sequences were extracted and blasted against the *R. ferrumequinum* genome, extended by 5 kb and blasted against a database containing all viruses as before. In another 'top-down' approach (blue box), the Premonition classified reads were mapped to the *R. ferrumequinum* genome first, and micro-assemblies generated that were blasted against the 'All virus' database either directly or after being extended by 5 kb on both sides. AVDB, all viruses database; MVDB, mammalian virus database; NMVDB, non-mammalian virus database; VH, viral hits; gVH, viral hits with homology to the bat genome; fgVH, viral hits using flanked genomic regions; RhiFer, *Rhinolophus ferrumequinum*; kb, kilobase pairs.

In a separate study, the Premonition BMM pipeline was applied in resolve viral taxa in febrile Nigerian patients resulting in a publication; Oguzie et al., Nature Communications. 2023 Aug 4;14(1):4693. see supplementary figure 2).

4. Is it [the Premonition Metagenomic pipeline] being introduced in this work?

Author response: Correct- this manuscript provides the first detailed description of the Premonition BMM pipeline as applied to an open-source dataset. The full algorithmic details can be found in Supplementary Information file; 'Generative Model' (pg. 2)

5. A search of the Microsoft website and literature did not yield any deeper technical details than what was included in the short methods section. I appreciate this succinctness but more details are needed. For example, what priors were used in the BMM? If data quality is poor (contamination) even so-called uninformative priors could have major effects on the posterior. If the data collection, storage, processing approaches introduced many sources of bias and/or error as stated in the conclusion, then the data would be very noisy so the priors and model assumptions are important to state explicitly.

We have included new supplementary information to explain these key points- see 'Supplementary Information, pg. 2; 'Generative Model'). To summarize, this section shows how the model accounts for issues such as reference contamination, variations in per-taxa read abundance within a degraded sample, sequencing error rates – while still supporting uniform priors on possible taxa. The generative model can accommodate sample-specific priors on taxa, though the results presented here do not require this. More generally, the approach we take is that BMM should reflect the abundance of taxa within a sample, and we view BMM as providing a correct result, it does reflect sample contamination, degradation, etc.

6. Was a multivariate normal distribution model assumed for the data?

Author's response: The model is a multinomial mixture model, but not a multivariate gaussian mixture model. Please see Supplementary Information for a detailed description in the form of a generative model (pg. 2)

7. Is the method completely unsupervised?

Author's response: This method is not completely unsupervised because some model parameters were fit using labeled/ground-truthed metagenomic datasets and ground-truthed simulated data.

8. Were the metagenomic reads only assigned to taxa or were any functional distributions inferred (genes, gene families, pathways)?

Author's response: We thank the reviewer for their inquiry. Reads were taxonomically assigned to the species-level. For this analysis, the BMM pipeline was not designed to resolve individual genes, gene families or genetic pathways.

9. Was taxonomic assignment benchmarked by comparison to any other conventionally used pipeline (e.g., Kraken2+Bracken, MetaPhlan (newest version), BLAST)?

Authors' response: We thank the reviewer for their inquiry. In response to this comment, we compared the performance of the Premonition BMM metagenomics pipeline against Kraken2 (Lu et al., Nat Protoc. 2022 (12):2815-2839), another commonly used pipeline to resolve one of the more “complex” specimens identified through the Premonition BMM analysis. Sample ERS248730 was identified to contain a large number of reads that corresponded to both an ungulate bloodmeal vertebrate host and an ungulate erythroparvovirus-1. The results from both analyses were in close agreement:

Premonition BMM pipeline output for sample ERS248730 in Krona plot format:

1. Mosquito & bloodmeal contents:

Premonition BMM pipeline output for sample ERS248730 in Krona plot format:

2. Viruses (0.02% of total aligned reads):

Kraken2 pipeline output for sample ERS248730 in Krona plot format:

<https://github.com/marbl/Krona/wiki>

The results are quite similar with regard to the resolved composition of the mixture. BMM performs the nucleic acid alignment as well as Kraken2, however the whole (BMM/Integrator) pipeline analysis goes beyond nucleic acid alignment to include coverage of reads and amino acid translation corresponding to the predicted contents of the sample (not shown). The largest challenge with Kraken was the preparation of a reference database. The prebuilt Kraken2 databases available online do not contain mosquito and mammalian references found in the reference databases used by the Premonition BMM pipeline (RefSeq and Genbank). After building the Kraken2 reference database (based on the same FASTA sequences used with the Premonition pipeline) 99% of the reads were found to align. It took approximately 10 days to build the database and required a virtual machine with 11 terabytes of memory. Such a computation would not have been possible in 2018 when we performed our first analysis with the Premonition BMM pipeline. The current databases have grown 6x since the original analysis

was performed. The largest machines available at that time had only about 4 terabytes of memory. The resulting Kraken database file is about 1 terabyte in size for the single sample.

10. Are there added benefits of this model approach over other classification methods?

Author's response: The primary benefits of the Premonition pipeline are scalability, speed of analysis and ability to cross-reference multiple large genomic sequence databases to make probabilistic inferences.

11. Is the focus of this work to highlight the Premonition BMM as employed with an interesting real-world dataset? Or, are biological insights into the Ag1000G genomes the main result?

Authors' response: The focus of this manuscript was to provide metagenomic insights into the publicly available Ag1000G dataset.

Reviewer #1 Specific comments:

12. Lines 60-62: Is there an implication here that the method used was able to determine pathogen-host connections? The blood meal is a mixture of DNA so how is a pathogen inferred to come from a particular unknown host?

Authors' response: We thank the reviewer for their clarifying question. The statement in lines 60-62 affirms the assertion that pathogen-host relationships may be inferred at least in part, through the application of metagenomic analyses of blood fed arthropods. We concur with the reviewer that blood meals may contain a mixture of vertebrate DNA and corresponding microbiota at the time of feeding, including potential pathogens. Further, the imbibing of incomplete blood meals from multiple hosts within the same gonotrophic cycle leads to mixtures of remnant vertebrate host DNA and their associated microbiota. In this study, we observed specimens containing reads assigned with high coverage to individual vertebrate hosts (Canid, Porcine, Equine, Hominid etc.) as well as several specimens containing reads assigned to multiple hosts (Fig.2). Additionally, we observed a single specimen (ERS248730) which contained many reads associated with an ungulate host and ungulate erythroparvovirus-1 which supports the assertion that host-pathogen associations may be resolved using metagenomic analyses. In another example, human hepatitis B virus (HBV) reads were observed in fifteen specimens containing a large number of human reads. HBV is not known to replicate in mosquitoes and therefore the most parsimonious explanation is that these viruses were present

in a blood meal taken from a human host. While not definitive, these findings suggest that metagenomic approaches can offer important inferences regarding host-pathogen associations that may be in turn, be verified through direct experimental work.

13. Line 81: What is meant by efficiently? Is there a relevant comparison to remark?

Authors' response: We thank the reviewer for their question. We have removed the word "efficiently" as there isn't a direct comparison we can draw at this time.

14. Line 90, 326-329: Can borderline cases, where posterior probabilities for two different taxa assignments were similar (e.g., 0.49 and 0.51), be quantified? For example, how many samples had borderline percentages and were not confirmed hybrids (middle ranges of points in Fig 1 cluster). It's hard to tell how many samples had borderline assignments but were not hybrid, which indicates uncertainty in the model.

Authors' response: At the original time of analysis, the three genomes available for reference of these taxa as represented in Fig. 1 are *An. gambiae*, *An. coluzzii* and *An. gambiae* PEST strain (a putative laboratory hybrid between what are now recognized as *An. gambiae* and *An. coluzzii*). Biologically, *An. gambiae* and *An. coluzzii* are reproductively compatible and 'hybrids' are easily generated in the lab and have been detected from African field sites. The genomes between these taxa are extremely similar and the uncertainty/inaccuracy is likely due to SNPs. The original publication of the Ag1000 data reported that there were no diagnostic SNPs to define these three taxa categories and further exploration of this is beyond the scope of this manuscript and has been more appropriately conducted by the *Anopheles gambiae* Genome Consortium. In response to the question, the model itself is probabilistic, so therefore the 'hybrids' detected by our analysis are likely on the genomic continuum between the two parental forms. We refer the reviewer to supplementary figure S2 for values of accuracy on these calls.

15. Lines 87-90, 102-104: Can this information be combined?

Authors' response: We thank the reviewer for the excellent suggestion. Lines 102-104 have been combined with lines 87-90 of the original draft manuscript. Lines 93-96 in the revised manuscript now reads: "An exceedingly small proportion of reads were assigned to plants, fungi, and other taxa. Less than one percent of all reads (0.861%; with an edit distance 20 or better) failed alignment to any sequence present in sequence databases at the time of analysis"

16. Line 111: Unclear which taxonomic level is being specified if not species (identified in the section title and next sentence?)

Authors' response: We thank the reviewer for the clarification. Lines 109-113 now read: “We evaluated BMM assignments of samples that were morphologically and genomically verified by the Ag1000G project, to members of the *An. gambiae* species complex, which are evolutionarily similar. On average, the *An. gambiae* species complex represented 93.3% of the probability mass given to Arthropod-assigned reads, with the remaining mass scattered across other anophelines.”

17. Lines 106-115: *Is this saying the method was validated by apriori verified sample identities compared to the model assignment probabilities? If so, how?*

Authors' response: Yes (see figure 1). All specimens were morphologically identified prior to sequencing as reported in the original publications of the Ag1000G datasets (see references 1 and 2).

18. Lines 131-135 Results and Discussion section: *These statements are a bit confusing about the difference between the way '99% identity' and 25% coverage' terms are being used for these 'other' vertebrate associated samples. Please clarify. Is the 99% identity referring to the BMM (MS Premonition) classification or an independent pipeline classifier for taxa assignment? What 132 genomes are being referred to? I am guessing that 99% identity means percent of reads assigned, whereas coverage is used for specific taxon genomes.*

Authors' response: We appreciate the reviewer's request for clarification. Both the 99% (read) identity and coverage statistics were computed by the BMM pipeline. We have rewritten Lines 136-145 in the revised manuscript to address the reviewer's questions more clearly. Lines 136-145 now read: “In contrast, the reads assigned to other vertebrates likely constitute evidence of blood feeding on non-human hosts. These signals represent over 613 million reads, sharing greater than 99% identity with reference genomes. We suggest this indicates blood feeding by these mosquitoes had occurred on singular and in some cases, multiple hosts. A total of 162 mosquito specimens contained non-mosquito reads corresponding to at least one chordate genome. 138 specimens contained human reads and 24 specimens contained non-human vertebrate reads. Blood meal hosts were assigned where vertebrate reads corresponded to at least 25% coverage of a reference genome sequence (Table 3).”

19. Lines 339-347 (Methods section): *Please define percent genome identity versus coverage separately. Both are referred to, but it is only stated they are calculated using the set of all reads with genome posterior probability greater than or equal to epsilon.*

Can the coverage vs identity difference be explained with an example for a specific taxon?

Authors' response: We thank the reviewer for their suggestion. Lines 369-373 in the revised text now read: "We extend the standard definitions of **genome percent identity** (i.e. the average percent identity of all reads assigned to a genome) and of **genome coverage** (i.e. the total fraction of genome locations for which at least one read is assigned to that location) to the BMM setting, where reads are not assigned with 100% probability." In the Supplementary Information we have included more nuanced definitions for additional mathematical background. In the 'Generative model' section of the Supplementary Information (pg. 2), read identity can be formally seen as related to edit distance. And in the 'Partitioned read dispersion' section of the Supplementary Information (pg. 6) coverage is related to limit as the number of genome partitions goes to the genome size.

17. There are multiple results in the Plasmodium results section. Can these be organized into separate paragraphs that start with the main point? Although I agree with a combined Results and Discussion for this work there is a lot of 'jumping around' from point to point within the section that reads in a jumbled way.

Authors' response: We thank the reviewer for the excellent suggestion. We have modified the Plasmodium Results section to reflect separate paragraphs with headings that capture the specific result reported. These include "***P. falciparum* Core and Apicoplast genome coverage:**", "***Plasmodium* read validation:**", and "**Positive Threshold for presence of *P. falciparum*:**"

18. Line 301: Please show how these posterior probabilities are being calculated as an explicit expression to obtain the distribution over samples of taxa assignment.

Authors' response: We thank the reviewer for the suggestion and point them to the 'Generative Model' section of the Supplementary Information for a detailed description of the mathematical expression by which posterior probabilities are calculated.

19. Lines 304-306: This is an interesting point and could potentially be generalized to determine environmental constraints (e.g., abiotic filtering, niche partitioning) affecting species distributions.

Authors' response: We thank the reviewer for their comment. We agree and believe that as sequencing technologies continue to improve and costs drop, metagenomic analyses of

arthropod microbiota will be more widely adopted. Further, we believe the application of metagenomic pipelines offers great promise in elucidating biological and ecological factors that dynamically influence mosquito vector populations.

20. Lines 312-313: Please provide a reference to the full description of the pipeline.

Authors' response: We thank the reviewer for their request. This manuscript represents the first publication containing a description of the pipeline architecture. We refer the reviewer to Supplemental Figure S1 for an illustrative description of the pipeline as well as the newly added 'Generative Model' description in the Supplementary Information (pg. 2) for a detailed mathematical description of the pipeline model.

21. Line 314: Was this done using Gibbs sampling?

Authors' response: No.

22. Lines 313-315: This is vague. "Based on statistical evidence" What statistic? Why? How does the pipeline do this?

Authors' response: We thank the reviewer for their inquiry. We explicitly describe the algorithmic details of the BMM in a newly added section of the Supplementary Information, labelled 'Generative Model'. This sentence has been rewritten to state: "The resulting probability distribution is a *Bayesian mixture model* (BMM), which is described in detail in the Generative Model section of the Supplementary Information." (Revised manuscript lines 341-343).

23. Line 317: Missing word "might have come [from] other anopheline species"

Authors' response: We thank the reviewer for the helpful suggestion. The insertion of [from] has been made in line 345 of the revised manuscript.

24. Line 329: Can the modelling assumptions be explicitly stated somewhere?

Authors' response: The model assumptions are now listed in the 'Generative Model' section of the Supplementary Information (pg. 6).

25. Line 346: Number 'of' reads vice "number or reads"?

Authors' response: We thank the reviewer for the request for clarification. Lines 346-47 now correctly reads as: " $E_{reads}[Anopheles]$ gives the expected number of reads contributed from non-anopheline dipterans."

Reviewer #2 (Remarks to the Author):

Reviewer #2 Overall comments (in italics):

Dr. Pastusiak and colleagues used a Microsoft Premonition Bayesian mixed model based (BMM) metagenomics pipeline to analyze the publicly available Ag1000G data and reported a number of interesting findings regarding species identification, hosts/Plasmodium/viruses/bacteria identification, and possible contaminations. I have a few major concerns that need to be addressed before the manuscript can be considered for publication.

Authors' response: We thank the reviewer for their thorough review of the manuscript. We endeavor to address the reviewer's concerns in the following responses.

Reviewer #2 Specific comments:

1. The description of the pipeline is not sufficient to help ensure reproducibility or rigor. For example, I could not find what alignment program was used in the pipeline.

Authors' response: We thank the reviewer for their comment. We have revised the manuscript to include a detailed mathematical description of the pipeline in the 'Generative Model' section of the Supplementary Information starting on page 2.

2. It is also not clear how the implementation of the pipeline significantly improved the efficiency or accuracy of the metagenomics analysis.

Authors' response: We thank the reviewer for their request for clarification on this point. To address this point, we compared the speed and accuracy of the Premonition BMM pipeline against another established metagenomic pipeline, Kraken2 to resolve the gut contents of a blood fed mosquito- ERS 248730. For additional details we refer the reviewer to our response to Reviewer #1's Comment #9.

3. The authors should further explore the causes of the uncertainty/inaccuracy associated with species assignment (lines 105-122). I don't fully understand Figure 1.

Authors' response: We thank the reviewer for their comment. At the original time of analysis, the three genomes for reference of these taxa as represented in figure 1 are *An. gambiae*, *An. coluzzii* and *An. gambiae* PEST strain (a putative laboratory hybrid between what are now recognized as *An. gambiae* and *An. coluzzii*). Biologically, *An. gambiae* and *An. coluzzii* are reproductively compatible and 'hybrids' are easily generated in the lab and have been detected

in Africa. The genomes between these taxa are extremely similar and the uncertainty/inaccuracy is likely due to SNPs. The original report on the AG1000 data reported that there were no diagnostic SNPs to define these three taxa categories and further exploration of this is beyond the scope of this manuscript and has been more appropriately conducted by the *Anopheles gambiae* Genome Consortium. In response to the question, the model itself is probabilistic, so therefore the 'hybrids' detected by our analysis are likely on the genomic continuum between the two parental forms. We refer the reviewer to Figure S2 for values of accuracy on these calls.

4. How many individuals were identified as males according to sample morphology? 84 male *Anopheles coluzzii* (formerly *An. gambiae* Mopti form) were identified in the metadata by the Ag1000G Consortium. There was no male that we could associate with a blood meal.

I suggest that the authors perform molecular identification of mosquito sex using known Y-linked markers (Hall et al., 2016, PNAS). Males could serve as a good control for vertebrate contamination as they do not feed on blood. It may also be interesting to compare/contrast the microbiome of the two sexes.

Authors' response: We have cross-verified the metadata provided for all sequenced specimens including those specimens' containing reads assigned to vertebrate hosts. No male mosquito specimens were identified as containing reads assigned to vertebrates except for the trace human DNA contamination found in all specimens which we note was likely introduced during collection of specimens and/or post-collection processing. We appreciate the reviewer's comment regarding comparing the microbiome of both sexes of mosquitoes. We agree this would be a worthwhile endeavor and could be done as part of a broader analysis of the Ag1000 dataset, however in this study, our primary focus was on the epidemiologically relevant sex, female *Anopheles gambiae* mosquitoes.

5. Lines 193-194, please provide support for the statement.

Authors' response: We thank the reviewer for their request for additional supporting text. The evidence for labeling taxa as contaminant is summarized in Column G of Table S2. Many of these are well-known contaminants present in many metagenomic studies as referenced in Cantalupo & Pipas (2019). Current opinion in virology, 39, 41-48. In each of these cases a genome coverage map was generated. In many contaminant taxa, sequences map to a single genome feature that is commonly used in vectors. For example, we detected human CMV in some samples, but all the sequence reads mapped to the immediate early promoter. SV40 was detected in some samples with all sequence reads covering the polyadenylation signal.

Similarly, we found sequences aligning to human adenovirus and avian leukosis virus, but in each case the alignments were to vector associated features. Finally, we found viral sequences that were present due to flow cell contamination. These included HIV and Influenza. These analyses are described in the text.

Reviewer #3 Overall Remarks:

The authors leverage data from the Anopheles 1000 genome project to develop a new metagenomics Bayesian mixed model within the Microsoft Premonition pipeline. Using this approach, they make a few key findings. Specifically, Anopheles seem to feeding at higher rates on non-human animals than previously thought with ~25% of feeds deriving from domestic and peri-domestic animals. Another interesting finding to emerge from this dataset was the surprisingly low amounts of fungal reads associated with the samples suggesting that fungus do not make up a substantial part of the mosquito microbiome. The finding of vertebrate viruses in the samples is really cool and supports earlier work using xenosurveillance. However, this is a large analysis with no experimental validation of the pipeline.

Authors' response: We thank the reviewer for their thorough review and summary comments.

Reviewer #3 Specific comments:

1. Without a more detailed explanation of the Microsoft Premonition pipeline, the authors need to go above and beyond to demonstrate the validity of their model. This can be achieved a number of ways including performing some experiments in the lab with mosquitoes known and unknown content.

Authors' response: We thank the reviewer for their comment. We have added a detailed description of the mathematical underpinning of the BMM pipeline in the Supplementary Information (see 'Generative Model'). With respect to demonstrating the validity of the model we refer Reviewer 3 to our response to Reviewer #1's Question #9 which details a head-to-head comparison of the BMM pipeline with another commonly used metagenomic pipeline Kraken2. The two pipelines were tasked with resolving one of the more "complex" specimens initially identified through the Premonition BMM analysis. Both analyses found an identical composition of reads that corresponded to both an ungulate bloodmeal vertebrate host and an ungulate erythroparvovirus-1. Additionally, we refer the reviewer to a recent publication- Dejosez et al. Cell. 2023 Mar 2;186(5):957-74 in which Kraken2 and Premonition BMM pipeline were used by

another academic group to perform comparative metagenomic analyses of endogenous bat viruses in a novel induced pluripotent stem cell model. Kraken2 and Premonition BMM pipeline analyses are visualized in supplementary figure S5. In a separate study, the Premonition BMM pipeline was applied to resolve viral taxa in febrile Nigerian patients resulting in a publication; Oguzie et al., Nature Communications. 2023 Aug 4;14(1):4693 (see supplementary figure 2).

2. I find it curious that bacteria were not found in all of the samples whether it be the actual microbiome or contaminating bacteria considering these were collected in the field.

Authors' response: We thank the reviewer for their observation. All samples contained bacterial reads (Fig. 6A). We established a threshold to identify samples containing high bacterial content (revised manuscript; lines 252-256). The low number of reads assigned to bacteria in some of the specimens may be due to variable and/or inadequate storage conditions or contamination at the time of collection or nucleic acid extraction. Mosquitoes were collected by a consortium of investigators conducting field studies across Africa for a wide variety of study objectives so there was not a standardized collection and preservation schema applied to all specimens prior to NGS sequencing. Therefore, one possible explanation for the low number of microbiota reads in some specimens may be due to specimen degradation resulting in failure to extract and sequence remnant DNA from specimens that were not well preserved prior to sequencing. This is noted in lines 302-309 of the revised manuscript.

3. Also, 478 samples had bacterial reads and 485 had plasmodium reads. Its curious that both, presumably unlinked habitants, would be found in roughly the same number of samples. Is this an artifact of the quality of sequencing data from those samples compared to the rest?

Authors' response: We thank the reviewer for their observation. We believe similar numbers of samples containing *Plasmodium* and bacterial reads (based on the cutoff criteria described in the Methods section) represent an experimental result and not an artifact of sequencing quality. The presence of bacteria and plasmodium were determined by multiple methods, of which sequence alignment by BMM was just one. As described in the text and methods, we generated genome coverage maps and examined gene organization before concluding bacterial or plasmodium presence (Fig 6b & Table S2). In addition, we ran our own quality controls and which led us to the conclusion that the observed bacterial and plasmodium reads were correctly assigned and not an artifact of sequencing.

Without sequence quality assessments and metadata, the data is purely observational and does not actually tell us much about the metagenome of mosquitoes. Consequently, this manuscript is principally about a pipeline; a pipeline that is protected by a proprietary cloud service.

Authors' response: We appreciate the reviewer's comment and would like to point out that the metadata for all specimens analyzed is freely available on the MalariaGEN/Ag1000G project website- Ag1000G phase 2 AR1 data release | MalariaGEN. We respectfully disagree that the manuscript is primarily describing the BMM pipeline. The BMM pipeline is simply the analytical tool which we used to explore and analyze an independently generated and complex genomic dataset. This study utilized the BMM pipeline to make novel biological observations regarding the Ag1000G dataset, rather than provide a detailed description of a proprietary pipeline.

Reviewers' comments:

Reviewer #1 (Remarks to the Author):

The authors have done a good job of responding to the first round of review comments. They provided necessary details that were absent about the main methodology used and included new subsections within supplementary information (SI) critical to the stated purpose of their research objective. Statistical method details are often relegated to supplemental materials and overlooked by readers. This is not surprising given the recent frenetic pace of scholarship but is fundamentally bad for science. It puts greater onus on authors and multiple reviewers to be cautious and require clarity in exposition of underlying statistical assumptions/models in SI. The authors' time and effort to explain these details is noticed and appreciated. Thank you.

Since most review responses were addressed via the new "Generative Model" and "Comparison of Premonition BMM and Kraken2" sections, the following comments focus primarily on SI:

- Line 105,113: Please describe details of "many simulated and actual metagenomic datasets" How were simulations generated? Major settings like $\lambda^{\text{effective}}$, $p_{\text{eff}}=4$, and $\gamma^{\text{miss}}=0.1$ were determined by "sweeps of simulated and actual metagenomic data" so this seems to be an important point for a first introduction of the Premonition BMM . Were properties of simulated datasets varied (e.g. SNR, numbers of samples, true taxa (species), genome database coverages)?

- Table S7 seems to be missing, or at least it doesn't appear in the files available to me on the manuscript tracking system. Table S7 is described in SI as containing a "Summary of samples containing potential novel bacteriophage" (see comment regarding line 227 below)

- Lines 49, 59-61, and 121-122: The authors explain that only the genome mixture probabilities were of interest and the other model parameters were treated as fixed. Thus, partitions and positions were given equal weight with Uniform priors. It seems likely the authors considered using Dirichlet priors, (conjugate to the multinomial distribution) in a hierarchical level like a form of Latent Dirichlet Allocation (LDA) for the reference genomes, but instead decided to use the special case of uniform priors. Why was major setting 1 of uniform priors decided?

I was asked to conduct a second round review of reviewer 3 (R3) specific comment 2 and 3 responses. Reviewer (R3) comment responses:

- Specific comment 2: Is the reference given in the response revision correct? "Therefore, we also examined the data following removal of taxa commonly associated with contamination[38]." Should this be reference 40?

-Specific comment 3: What are these other quality controls referred to in the additional statement "we ran our own quality controls and which led us to the conclusion that the observed bacterial and plasmidium reads were correctly assigned and not an artifact of sequencing."

Other minor points:

- Lines 242-244: Is the rationale behind this sample exclusion that samples with contamination may have lots of reads but that does not resolve into more taxonomic families compared to samples of a similar environment without contamination. Basically, contamination means more reads but not necessarily more families? Were any other cutoffs tried?

- Line 227: where is the determination of which bacterial sequences were present as potential hosts for phages? Is this reported in column D of table S7? I do not see a file containing Table S7. Perhaps it was mislabeled or I missed it.

- Fig S3: I don't see the brown or light brown circles described in the caption. There are only three

colors (yellow, blue, and orange). These krona plots are also not very useful to look at in this format for comparison.

- Line 232: Was something forgotten here, it seems to be a sentence fragment? "Avian leukosis virus sequences"

- Line 232: A sentence or two minimum should be added to introduce Pickaxe and what it does with the appropriate Cantalupo et al., reference. This could be similar to how Integrator was introduced on lines 222,223 of the main text and does not necessarily need to be a whole paragraph outline. But something should be said.

- Line 143: I don't see "not clean" taxa defined anywhere? Does this refer to known contaminants?

- Line 96: The meaning of the last sentence is mysterious. What correction? I read this paragraph as explaining how the (big) assumption of unbiased sequencing biases the model towards shorter genomes since a smaller length λ of smaller genomes would produce a larger value of $1/\lambda$. I'm guessing this correction is referring to the $\lambda^{\text{effective}}$ introduced in the next two paragraphs.

- Line 94: What is the meaning of higher order reference genomes? Higher taxonomic rank?

- Line 74: Is this saying that genome g_i at position j contributed nucleotide at position k of read x (namely the nucleotide x_k referenced in lines 36-40)?

- Line 28: Using an inconsistent index for nucleotide position in a genome or read ('j' compared to line 37 'k') is somewhat confusing but I assume the authors thought it would be obvious from context.

- Line 25,37 vs Main text line 312: different notation used for collections of sample reads $S=\{r_1, \dots, r_m\}$ vs X in SI

Reviewer #2 (Remarks to the Author):

The authors have addressed my concerns and I recommend acceptance of this manuscript. Just a few minor suggestions: 1) line 21, *Anopheles gambiae* should be *An. gambiae*; 2) line 108, what are the other anophelines and why are 6.7% of arthropod-assigned reads not assigned to the *gambiae* complex? 3) line 570, shouldn't it be Communications Biology?

RESPONSE TO REVIEWERS' COMMENTS

(2nd revision submission)

Reviewer #1 Remarks to the Author: *“The authors have done a good job of responding to the first round of review comments. They provided necessary details that were absent about the main methodology used and included new subsections within supplementary information (SI) critical to the stated purpose of their research objective. Statistical method details are often relegated to supplemental materials and overlooked by readers. This is not surprising given the recent frenetic pace of scholarship but is fundamentally bad for science. It puts greater onus on authors and multiple reviewers to be cautious and require clarity in exposition of underlying statistical assumptions/models in SI. The authors' time and effort to explain these details is noticed and appreciated. Thank you.”*

Authors' response: We thank the reviewer for their thorough review of the revised manuscript and supplementary information. We acknowledge the reviewer's comments regarding inclusion of statistical methods in supplementary materials and hope we have satisfied the reviewer's request for a more detailed description of the underlying model and assumptions in the latest revisions. We greatly appreciate the reviewer's attention to detail and believe the quality of the manuscript has been substantially improved as we result of the reviewer's useful comments and suggestions. Thank you.

Reviewer 1 Overall comments: “*Since most review responses were addressed via the new "Generative Model" and "Comparison of Premonition BMM and Kraken2" sections, the following comments focus primarily on SI:*”

1. SI Line 105,113: “*Please describe details of "many simulated and actual metagenomic datasets". How were simulations generated? Major settings like $\lambda^{\text{effective}}$, $p_{\text{eff}}=4$, and $\gamma^{\text{miss}}=0.1$ were determined by "sweeps of simulated and actual metagenomic data" so this seems to be an important point for a first introduction of the Premonition BMM. Were properties of simulated datasets varied (e.g. SNR, numbers of samples, true taxa (species), genome database coverages)?”*

Authors' response: We thank the reviewer for their inquiry. The Premonition pipeline has been validated with various synthetic (simulated) runs. We constructed multiple artificial mixtures of reads and validated that these mixtures can be recovered with BMM. Artificial mixtures were specified in a domain-specific language that allowed mixture weights to be specified per taxon, so realistic abundances of e.g. host nucleic acids and viral RNA could be modeled. For example, a synthetic mixture could include 90% reads drawn from a mosquito reference, 9% from human, ~1% from Wolbachia bacteria and just 0.01% reads drawn from references of Zika virus. Reads were drawn at random using the ART sequencing read simulator (Huang, W., Li, L., Myers, J.R., & Marth, G.T. ART: a next-generation sequencing read simulator. *Bioinformatics*. **28**, 593-594 (2012)). We used various simulated read error rates from 1% to 10%. AUC was computed by applying the MAP (*maximum a posteriori*) rule to each read, meaning the

taxon with highest probability of contributing that read according to BMM was chosen for counting positive and negative outputs. In the first mode, the pipeline was challenged to reconstruct mixtures assuming the pipeline had knowledge of the genomic references for species in the test mixture. In this mode, BMM was ~99.99% accurate to fully reconstruct a mixture at the read level despite asymmetric abundances of taxa. In the second mode, we assumed all species in the test mixture were novel (e.g. even hosts such as *Homo sapiens* would also be simulated as a novel species) by blinding BMM from accessing any references of the test taxa, and then evaluated accuracy of correctly reconstructing mixtures at the genus and family levels. Accuracies were >84% at placing novel reads in the correct genus, >97% at placing novel viruses in the correct family, and >94% at placing novel chordates in the correct family, with results varying significantly across clades depending on genomic representation in reference databases. In addition, biological samples with known ground truths were also analyzed in this fashion based on proprietary samples prepared by the authors (e.g. mosquito RNA samples spiked with known titers of arthropod-borne (arbo) and non-arboviruses that were then sequenced). The complete parameter sweeping architecture is non-trivial and requires cloud-scale infrastructure, which we intend to expand on in a future publication. The citation for ART was added in lines 105 and 114 of the revised Supplementary Information (Huang et al. 2012 [1]).

2. “Table S7 seems to be missing, or at least it doesn't appear in the files available to me on the manuscript tracking system. Table S7 is described in SI as

containing a ‘Summary of samples containing potential novel bacteriophage’ (see comment regarding line 227 below).’’

Authors’ response: The bacterial sequences in question are indeed reported in column D of Table S7 (now included). We greatly apologize for the omission of Table S7 from the revised submission- this was an oversight during the submission process. This has been remedied with its careful inclusion with the most recently submitted version of the manuscript, Supplementary Information and supporting tables and figures.

3. SI Lines 49, 59-61, and 121-122: *“The authors explain that only the genome mixture probabilities were of interest and the other model parameters were treated as fixed. Thus, partitions and positions were given equal weight with Uniform priors. It seems likely the authors considered using Dirichlet priors, (conjugate to the multinomial distribution) in a hierarchical level like a form of Latent Dirichlet Allocation (LDA) for the reference genomes, but instead decided to use the special case of uniform priors. Why was major setting 1 of uniform priors decided?”*

Authors’ response: This is an excellent point, and our approach can incorporate these priors (or others). The reason we used uniform priors was motivated by real-world usage patterns as opposed to computational efficiency concerns. In practice, users wish to take minimal assumptions about input datasets, whereas hyperparameters in prior distributions require users to have some knowledge / provide assumptions about species abundances in nature. However, current scientific knowledge of species

presence / absence, abundance, range, and dynamics is very limited, so users do not have good estimates for these hyperparameters. In the future, as these complementary datasets improve, we believe they will motivate the use of non-uniform priors as a disciplined approach to introduce other background data into BMM. On the other hand, users often do have a better understanding of how sequencing protocols might, *a priori*, influence the parts of a genome that are more likely to be observed. For example, host ribosomal RNA might be expected during viral metagenomic analyses or mitochondrial regions might be targeted for assessing animal presence / absence in environmental DNA. For these reasons, our model contains non-uniform priors for expressing this information at the genome level.

*****NOTE:** Reviewer #1 was asked to conduct a second round review of reviewer 3 (R3) specific comment 2 and 3 responses. These were not numbered and are included here:

- R3 Specific comment 2: *Is the reference given in the response revision correct? "Therefore, we also examined the data following removal of taxa commonly associated with contamination[38]." Should this be reference 40?*

Authors' response: We thank the reviewer for their careful review and suggestion. The reference number has been corrected in the revised manuscript to reflect the correct reference:

40. Zinter, M. S., Mayday, M. Y., Ryckman, K. K., Jelliffe-Pawlowski, L. L. & DeRisi, J. L. Towards precision quantification of contamination in metagenomic sequencing experiments. *Microbiome*. **7**, 1-5 (2019).

- R3 Specific comment 3: *refers to Response to Reviewer Comments_20240121 (included below for context): What are these other quality controls referred to in the additional statement "we ran our own quality controls, and which led us to the conclusion that the observed bacterial and plasmodium reads were correctly assigned and not an artifact of sequencing. "?*

Authors' response: We thank the reviewer for the follow-up question. The additional quality control measures employed were to examine the alignment scores and BMM probability outputs to confirm taxa calls.

Other minor points raised by Reviewer 1:

4. Manuscript Lines 242-244: *"Is the rationale behind this sample exclusion that samples with contamination may have lots of reads but that does not resolve into more taxonomic families compared to samples of a similar environment without contamination. Basically, contamination means more reads but not necessarily more families? Were any other cutoffs tried?"*

Authors' response: We thank the reviewer for their inquiry. The only cutoff or exclusion criteria applied was where the number of bacterial reads exceeded mosquito reads for a given specimen. Specimens containing more bacterial reads than mosquito reads were considered contaminated and therefore excluded from the analysis.

5. Manuscript Line 227: *"Where is the determination of which bacterial sequences were present as potential hosts for phages? Is this reported in column D of table*

S7? I do not see a file containing Table S7. Perhaps it was mislabeled or I missed it.”

Authors’ response: We greatly apologize for the omission of Table S7 from the revised submission- this was an oversight during the submission process. This has been remedied with its careful inclusion with the most recently submitted version of the manuscript, Supplementary Information and supporting tables and figures. The bacterial sequences in question are indeed reported in column D of Table S7 (now included).

6. SI Fig S3: “I don’t see the brown or light brown circles described in the caption. There are only three colors (yellow, blue, and orange). These krona plots are also not very useful to look at in this format for comparison.”

Authors’ response: We thank the reviewer for their astute observation. The only colors displayed are indeed yellow, blue, and orange. The color yellow is used to illustrate the HBV open reading frames encoding viral proteins Orange and blue indicate the sequence coverage depth and percent identity respectively. The intention of including individual Circos plots was to demonstrate coverage across the entire HBV genome for the 15 specimens displayed. The figure caption (revised SI lines 284-286) has been revised to state: **“Figure S3. HBV Coverage Maps. Panels A – O show the sequence coverage maps of HBV found in each of 15 samples. Yellow illustrates the HBV open reading frames encoding viral proteins. Orange and blue indicate the sequence coverage depth and percent identity respectively.”**

7. SI Line 232: “Was something forgotten here, it seems to be a sentence fragment? "Avian leukosis virus sequences"”

Authors’ response: We thank the reviewer for catching this errant sentence fragment. It has been removed from the text for clarity.

8. SI Line 232: *A sentence or two minimum should be added to introduce Pickaxe and what it does with the appropriate Cantalupo et al., reference. This could be similar to how Integrator was introduced on lines 222,223 of the main text and does not necessarily need to be a whole paragraph outline. But something should be said.*

Authors’ response: We thank the reviewer for their excellent suggestion. We have added an additional brief description of PickAxe and reference to the Supplementary Information. SI Lines 230-232 now read: “PickAxe is a virus detection pipeline that uses both nucleic acid identity and protein amino acid sequence identity to uncover known and novel viruses from metagenomic NGS data.[3]”

3. Cantalupo, P. G. *et al.* Raw sewage harbors diverse viral populations. *MBio*, **2**, 10.1128/mbio.00180-11 (2011).

9. SI Line 143: *I don't see "not clean" taxa defined anywhere? Does this refer to known contaminants?*

Authors’ response: We thank the reviewer for their question. “Not clean” does not refer to known contaminants but rather indicates reads that were assigned but did not pass the default PrinSeq and/or Cutadapt filters. This is indicated by the asterisk in

Table 1: (line 639 of the revised manuscript) ***Flagged and removed by PrinSeq or Cutadapt.**

10. SI Line 96: *The meaning of the last sentence is mysterious. What correction? I read this paragraph as explaining how the (big) assumption of unbiased sequencing biases the model towards shorter genomes since a smaller length λ of smaller genomes would produce a larger value of $1/\lambda$. I'm guessing this correction is referring to the $\lambda^{\text{effective}}$ introduced in the next two paragraphs.*

Authors' response: Thank you for pointing out that this sentence causes confusion. We have removed it. The intent of this statement was to point out that the $1/\lambda_i$ term (the “correction”) emerges naturally from setting partition weights according to a uniform sampling assumption. We find this mathematically interesting but agree that calling this “important” should be left to the reader to decide.

11. SI Line 94: *What is the meaning of higher order reference genomes? Higher taxonomic rank?*

Authors' response: We thank the reviewer for their clarifying question. “Higher order” in this case refers to eukaryotic organisms (compared to prokaryotes and viruses). We have changed the text to replace “higher order” with “eukaryotic”.

12. SI Lines 73-74: *Is this saying that genome g_i at position j contributed nucleotide at position k of read x (namely the nucleotide x_k referenced in lines 36-40)?*

Authors' response: We thank the reviewer for their question. We appreciate that this notation is confusing and have changed it. The intended reading is that \mathbf{x}_k represents (the entire) k^{th} read among the collection of all reads \mathbf{X} . We have rewritten the **Reads and read probabilities** Section (starting SI Line 37) so that c_q now represents the q^{th} character of a specific read \mathbf{x} , replacing the original notation x_q that can be easily confused with the bolded notation \mathbf{x}_k . The text in SI Lines 37-38 have been changed to read: "... is a vector $\mathbf{x} = (c_1, \dots, c_l)$ where each $c_q \in \Omega$ is the q^{th} character of read \mathbf{x} ." With this change we hope the intended reading of SI Lines 73-74 is clearer: ***read k starts from position j in genome g_i .***

13. SI Line 28: *Using an inconsistent index for nucleotide position in a genome or read ('j' compared to line 37 'k') is somewhat confusing but I assume the authors thought it would be obvious from context.*

Authors' response: We thank the reviewer for this point. We recognize there is the need to introduce many indices, and we have taken the approach of treating indices i, j, k as being lexically scoped, so an occurrence of index i in a later section of the text can range over a different set than in an earlier section. When starting a new lexical scope, we have tried to introduce indexing variables in alphabetical order, starting with i , to improve readability. The alternative would be to expand the set of identifiers considered to be indices, which may cause confusion with symbols usually associated with constants or parameters (e.g. a, b, c, d). We hope this is a satisfactory motivation for this notation.

14. SI Line 25,37 vs Main (manuscript) text line 312: *different notation used for collections of sample reads $S=\{r_1, \dots, r_m\}$ vs X in SI*

Authors' response: Thank you for pointing this out. We have updated the notation in the Materials and Methods section of the revised manuscript to match the notation in the Supplementary Information.

Reviewer #2 (Remarks to the Author): *The authors have addressed my concerns and I recommend acceptance of this manuscript. Just a few minor suggestions:*

1. Manuscript Line 21: *Anopheles gambiae* should be *An. gambiae*.

Authors' response: We thank the reviewer for catching this typo. Line 21 now reads: “sequencing methods to catalogue genetic diversity across African *An. gambiae*...”

2. Manuscript Line 107: *what are the other anophelines and why are 6.7% of arthropod-assigned reads not assigned to the gambiae complex?*

Authors' response: We thank the reviewer for their inquiry. Arthropod reads assigned to anophelines outside the *An. gambiae* species complex included: *An. funestus*, *An. maculatus*, *An. minimus*, *An. albimanus*. All taxa assignments made by BMM represent probabilistic estimates to the species-level based on homology with the reference sequences used at the time of analysis. The probability of BMM assignments is influenced by alignment of specimen reads across reference sequences of varying size, quality and homology as well as pre-processing errors introduced during sequencing. With genomic reference databases expanding in size and quality, we expect the

probabilistic uncertainties of BMM assignments to decrease resulting in improved accuracy of species-level assignments in future analyses.

3. Manuscript Line 570: *shouldn't it be Communications Biology?*

Authors' response: We thank the reviewer for catching this omission. Line 574 of the revised manuscript now reads: "...This does not alter our adherence to Communications Biology's..."

REVIEWERS' COMMENTS:

Reviewer #1 (Remarks to the Author):

My comments have all been adequately addressed in the revised manuscript.